# Direct methane protonic ceramic fuel cells with self-assembled Ni-Rh bimetallic catalyst

Kyungpyo Hong [1,7], Mingi Choi[2,7], Yonggyun Bae [1,3], Jihong Min[1], Jaeyeob Lee[4], Donguk Kim[4], Sehee Bang[4], Han-Koo Lee[5], Wonyoung Lee [4,6] & Jongsup Hong [1]

Direct methane protonic ceramic fuel cells are promising electrochemical devices that address the technical and economic challenges of conventional ceramic fuel cells. However, Ni, a catalyst of protonic ceramic fuel cells exhibits sluggish reaction kinetics for $CH_4$ conversion and a low tolerance against carbon-coking, limiting its wider applications. Herein, we introduce a self-assembled Ni-Rh bimetallic catalyst that exhibits a significantly high $CH_4$ conversion and carbon-coking tolerance. It enables direct methane protonic ceramic fuel cells to operate with a high maximum power density of ~0.50 W·cm$^{-2}$ at 500 °C, surpassing all other previously reported values from direct methane protonic ceramic fuel cells and even solid oxide fuel cells. Moreover, it allows stable operation with a degradation rate of 0.02%·h$^{-1}$ at 500 °C over 500 h, which is ~20-fold lower than that of conventional protonic ceramic fuel cells (0.4%·h$^{-1}$). High-resolution in-situ surface characterization techniques reveal that high-water interaction on the Ni-Rh surface facilitates the carbon cleaning process, enabling sustainable long-term operation.

Protonic ceramic fuel cells (PCFCs) are promising electrochemical devices with low operating temperatures (<600 °C), high electrochemical performance, high efficiency, and no fuel dilution[1–3]. Moreover, the high fuel flexibility of PCFC, which uses hydrocarbons, especially methane, instead of pure hydrogen, can be a breakthrough for wide-ranging applications[4–8]. Methane is a good candidate for alternative fuel due to its ~3-fold higher volumetric energy density (~10 MJ/L at 250 bar) than hydrogen (~3 MJ/L at 350 bar)[9]. It is also readily storable and transportable using existing infrastructures, such as tanks and gas pipelines, facilitating direct integration of PCFCs within established value chains. Therefore, one promising strategy is the development of direct methane PCFC, where methane serves as the fuel through a methane steam reforming reaction ($CH_4 + H_2O \rightarrow CO + 3H_2$, $\Delta H_{293K} = 206$ kJ/mol) at the fuel electrode,

reducing system size and complexity caused by the additional fuel reformer[10].

However, several challenges must be addressed for the direct methane PCFCs. First, conventional catalysts such as nickel (Ni) exhibit substantially reduced $CH_4$ reforming activity as the operating temperature is lowered, which induces insufficient hydrogen supply[11]. Carbon-coking, which causes significant performance degradation, is also a challenging problem for sustainable operation[12]. Therefore, to achieve the high performance and stability of direct methane PCFCs at low temperatures, structural and/or material modification of the fuel electrode is necessary to meet the following requirements: (1) highly active catalysts for $CH_4$ activation, (2) self-carbon cleaning properties to mitigate the carbon-coking, and (3) large and uniformly distributed catalyst for maximizing both $CH_4$ activation and self-carbon cleaning.

[1]School of Mechanical Engineering, Yonsei University, Seoul, Republic of Korea. [2]Department of Future Energy Convergence, Seoul National University of Science & Technology, Seoul, Republic of Korea. [3]Department of Zero-carbon Fuel & Power Generation, Korea Institute of Machinery & Materials, Daejeon, Republic of Korea. [4]School of Mechanical Engineering, Sungkyunkwan University (SKKU), Suwon, Republic of Korea. [5]Pohang Accelerator Laboratory, Pohang University of Science and Technology (POSTECH), Pohang, Republic of Korea. [6]SKKU Institute of Energy Science and Technology (SIEST), Sungkyunkwan University, Suwon, Republic of Korea. [7]These authors contributed equally: Kyungpyo Hong, Mingi Choi. ✉e-mail: leewy@skku.edu; jongsup.hong@yonsei.ac.kr

Various strategies such as multistep infiltration, pulsed laser deposition, and atomic layer deposition have been explored[13-15]. However, these are complicated, time-consuming, and cost-ineffective for large-scale applications due to various manufacturing process parameters.

The development of Ni-based bimetallic catalysts containing small amounts of noble metals, such as Rh, Ru, and Pd, provides an effective strategy to address these limitations. These bimetallic catalysts significantly improve the catalytic activity through the synergistic effect of two materials, facilitating $H_2$ spillover and enhancing carbon-coking tolerance[16-20]. Additionally, exsolved metal catalysts can provide an increase in gas conversion with evenly dispersed nanosized particles and enlarge the metal catalyst/support interface[4]. Anchored structure of exsolved particles at the interface between catalyst and catalyst support also ensures high structural stabilities without agglomeration even at elevated temperatures[21]. Therefore, taking advantage of both approaches—bimetallic catalyst and exsolved nanoparticles—is a rational approach for designing a reactive and robust fuel electrode for a direct methane electrochemical cell.

In this study, we report a direct methane PCFC with high performance and stability with a self-assembled Ni-Rh bimetallic catalyst. Deliberately fabricated Ni-diffused $BaZr_{0.4}Ce_{0.4}Y_{0.1}Yb_{0.1}O_{3-\delta}$ (BZCYYb) is utilized as a platform for Rh nanoparticles by one-step infiltration. Self-assembly between infiltrated Rh and diffused Ni is facilitated by subsequently exsolved Ni particles under reduction condition. PCFC with a self-assembled Ni-Rh bimetallic catalyst exhibits a significantly high performance of 1.13 $W\cdot cm^{-2}$ at 650 °C and 0.50 $W\cdot cm^{-2}$ at 500 °C under direct methane fuel conditions, surpassing other previously reported direct methane PCFCs and solid oxide fuel cells (SOFCs). It is attributed to significantly enhanced $CH_4$ conversion by the Ni-Rh bimetallic catalyst, achieving nearly thermodynamic equilibrium. Moreover, this catalyst shows outstanding electrochemical stability with a degradation rate of 0.02%·h$^{-1}$at 500 °C over 500 h, which is ~20-fold lower than that of the conventional PCFC (0.4%·h$^{-1}$). In-situ diffuse reflectance infrared Fourier transform spectroscopy (DRIFTS) and synchrotron-based in-situ X-ray photoelectron spectroscopy (XPS) measurements reveal that the high-water dissociation properties of the Ni-Rh bimetallic catalyst induce the self-carbon cleaning on the catalyst surface. Our approach, a self-assembled bimetallic catalyst, is readily simple and cost-effective, enabling the extensive application to other electrochemical cells that requires the reforming of other various gases such as hydrocarbon fuels and ammonia.

## Results

### Fuel cell structure and self-assembled Ni-Rh bimetallic catalyst

To architecture the fuel electrode with a self-assembled Ni-Rh bimetallic catalyst, we combine the exsolution and one-step infiltration processes on the Ni/BZCYYb anode-support single cell configuration. We first deliberately designed a Ni-diffused BZCYYb fuel electrode using the interdiffusion mechanism of Ni (See Supplementary Fig. 1) and utilized it as a platform for the subsequent self-assembly between exsolved Ni particles and infiltrated Rh particles to form bimetallic catalysts. As shown in Fig. 1a, we decorated the surface of Ni-diffused BZCYYb with Rh nanoparticles through the one-step infiltration process. Since Rh is highly miscible with Ni, the infiltrated Rh particles are autonomously mixed with subsequently exsolved Ni during $H_2$ reduction, resulting in a Ni-Rh bimetallic catalyst. Within the applicable temperature range in this study, we sintered the fuel electrode at 1500 °C, which demonstrates the largest grain size, to reduce the ohmic resistance and to facilitate the Ni-Rh bimetallic alloy formation through a large number of Ni exsolution (See Supplementary Fig. 2). We denoted the cell without Rh decoration and with Rh decoration as REF and Ni-Rh cell, respectively.

Figure 1b and c shows the SEM images of the Ni-Rh cell before and after reduction, respectively. In the Ni-Rh cell before reduction, Rh nanoparticles (4–8 nm) are decorated by infiltration on the BZCYYb

surface. XRD patterns and EDS mapping in Supplementary Fig. 3, Fig. 1d–1 and d-3 shows that Rh nanoparticles exist solely as a partially oxidized metallic phase (RhO and Rh, ~2.2 Å for Rh (111)[22]) without mixing with Ni inside the BZCYYb lattice before reduction. On the other hand, after reduction, Ni and Rh co-exist as a bimetallic alloy (~2.15 Å for Ni-Rh (111)[23]) with a particle size of 5–10 nm (See Fig. 1e-1, e-3). EDS mapping of other compositions are displayed in Supplementary Fig. 3. The anchored structure around the interface between the Ni-Rh bimetallic catalysts and BZCYYb support confirms that Ni overcoats Rh during the exsolution process and forms the Ni-Rh bimetallic alloy. Interestingly, the Ni-Rh bimetallic catalyst shows ~7-fold higher surface coverage of 85–87% and ~5-fold smaller particle size (8–11 nm) than those of exsolved Ni particles in the REF cell (Supplementary Fig. 1f). The higher surface coverage with the smaller particle size is attributed to the presence of Rh nanoparticles, providing additional nucleation sites for exsolution under the same amount of diffused Ni[24]. In addition, the smaller particle size of the Ni-Rh bimetallic catalyst is attributed to the higher surface energy of Rh (2828 $mJ/m^2$) than Ni (2364 $mJ/m^2$), preserving their particle size without agglomeration[25,26]. The smaller particle size maximizes the catalyst surface area and induces strong metal support interactions, increasing gas conversion and catalytic activity[27]. The particle size of the Ni-Rh bimetallic catalyst is substantially smaller or at least comparable to recently reported values through exsolution (~50 nm), multistep infiltration (~20 nm), and atomic layer deposition (~10 nm), demonstrating the feasibility of our simple approach for enlarging the catalytic active sites[13,15,28].

### Performance and electrochemical/thermochemical analyses

We evaluated the electrochemical performances of REF and Ni-Rh cells under hydrogen (97% $H_2$ and 3% $H_2O$) and methane ($H_2O/CH_4$) with S/C = 2 and S/C = 1, as shown in Fig. 2, Supplementary Figs. 4, 5, and Supplementary Table 1. The measured open circuit voltages (OCVs) under different partial pressures of $H_2$ ($P_{H2}$) are close to the theoretical values, confirming the sufficient gas tightness of the electrolyte, as shown in Supplementary Figs. 6, 7 and Supplementary Table 2 [29]. Under $H_2$ operation, the Ni-Rh cell demonstrates ~1.20 and ~1.06-fold higher maximum power densities (MPDs) of ~1.47 $W\cdot cm^{-2}$ at 650 °C and ~0.69 $W\cdot cm^{-2}$ at 500 °C than those of the REF cell (~1.22 $W\cdot cm^{-2}$ at 650 °C and ~0.65 $W\cdot cm^{-2}$ at 500 °C). Figure 2a–d shows that the improved electrochemical performances of the Ni-Rh cell are more evident with the $CH_4$ fuel and a lower operating temperature. Under $CH_4$ operation (S/C = 2), the Ni-Rh cell exhibits ~1.44-fold higher MPDs at 650 °C (~0.78 $W\cdot cm^{-2}$ for the REF cell and ~1.13 $W\cdot cm^{-2}$ for the Ni-Rh cell, respectively), and ~2-fold higher MPDs at 500 °C (~0.25 $W\cdot cm^{-2}$ for the REF cell and ~0.50 $W\cdot cm^{-2}$ for the Ni-Rh cell, respectively). These trends are also evident in the lower steam condition of S/C = 1 in Supplementary Note 1.

As shown in Fig. 2e and Supplementary Table 3, the Ni-Rh cell exhibits outstanding MPDs under $CH_4$ operation, outperforming previously reported values for PCFCs and SOFCs[4-7,30-34]. Specifically, the Ni-Rh cell shows a particularly high MPD at low temperatures, such as ~0.50 $W\cdot cm^{-2}$ at 500 °C under $CH_4$ operation of S/C = 2. These higher MPDs of the Ni-Rh cell are primarily attributed to the lowest area-specific polarization resistance ($ASR_{electrode}$), corresponding to the electrode resistance as shown in Fig. 2f. To further investigate the performance improvement in electrochemical reactions, we use electrochemical impedance spectroscopy (EIS) measurements and distributed relaxation time (DRT) analyses to deconvolute the $ASR_{electrode}$ into three distinct frequency ranges—high (>$10^3$ Hz), medium (10-$10^3$ Hz), and low (<10 Hz)—corresponding to the charge transfer at the triple phase boundary (TPB) of the fuel and air electrodes, the gas adsorption process and the overall surface reactions at the electrodes, and the gas diffusion and fuel reforming in the fuel electrode, respectively.[35,36]. Figure 2g shows the deconvoluted $ASR_{electrode}$ of the

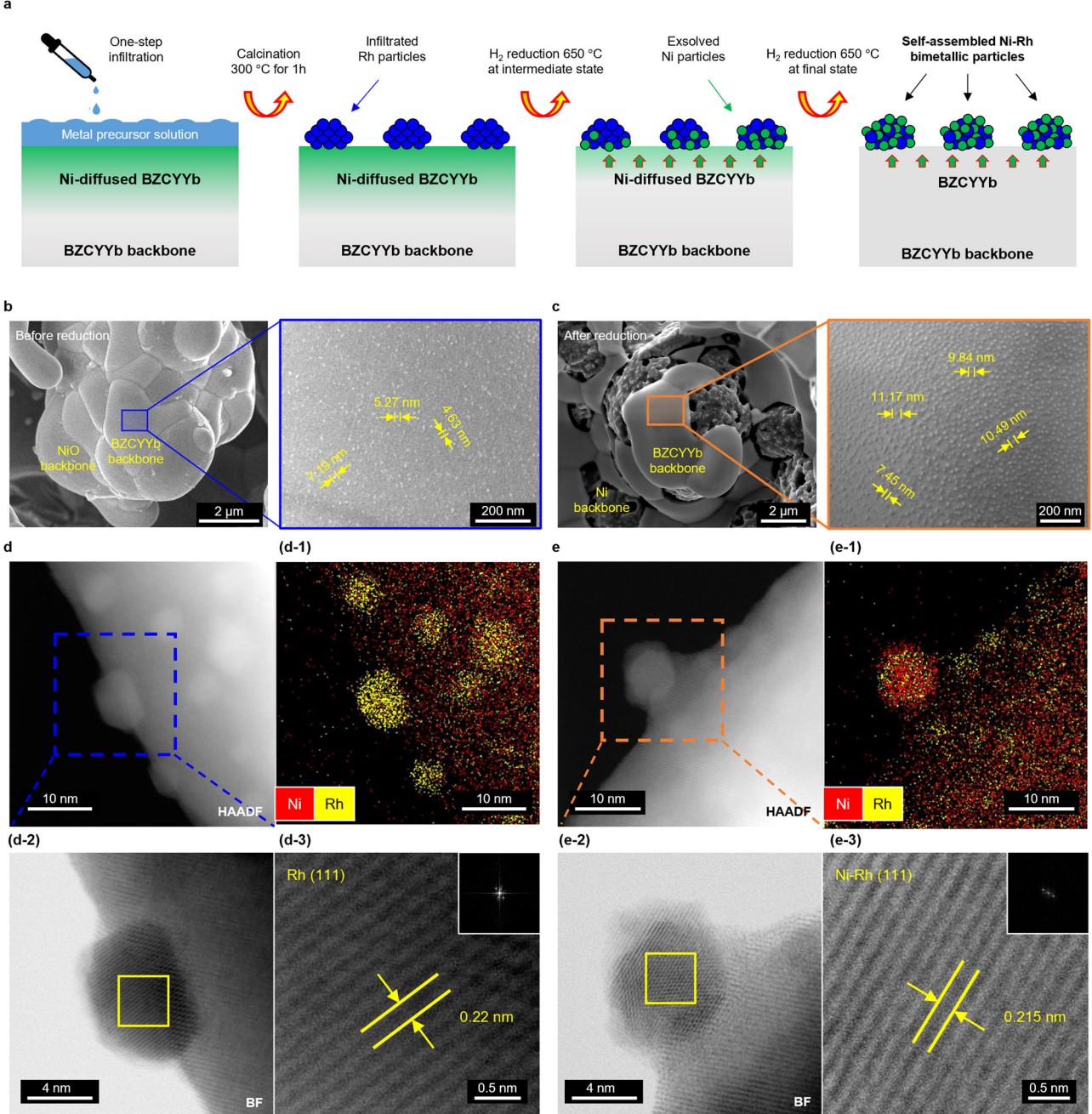

**Fig. 1 | Self-assembly process of the Ni-Rh bimetallic catalyst. a** Schematics of the self-assembly process between infiltrated Rh and exsolved Ni particles. High magnification SEM images of the fuel electrode morphology of Ni-Rh cell (**b**) before reduction and (**c**) after reduction, respectively. Structure and chemical composition of the catalyst on the fuel electrode by TEM and EDS mapping with lattice spacing images for the Ni-Rh cell (**d**) before reduction and (**e**) after reduction, respectively.

REF and Ni-Rh cells at 500 °C. At high and medium frequencies, the Ni-Rh cell exhibits slightly lower resistances than the REF cell under all fuel conditions. This is attributed to the high electrochemical activity of the Ni-Rh bimetallic catalyst for charge transfer at the TPB and the overall hydrogen oxidation reactions (HOR) at the fuel electrode compared to Ni[37]. When switching the fuel from $H_2$ to $CH_4$, the medium frequency resistances significantly increase by a similar magnitude in both REF and Ni-Rh cells due to the slow gas–solid interaction caused by the reduced partial pressure of $H_2$ and the sluggish $CH_4$ adsorption. However, although the low-frequency resistances for the REF cell significantly increase by sluggish gas reforming under $CH_4$ operation, those for the Ni-Rh cell almost remain unchanged. The same EIS trend

in symmetric cell analysis in Supplementary Fig. 8 further clarifies the effect of the Ni-Rh cell on the electrochemical results under $H_2$ and $CH_4/H_2O$ environment. Therefore, we can conclude that enhanced electrochemical performance of the Ni-Rh cell is predominantly attributed to the fuel electrode performance since other components such as electrolyte and cathode are all identical between REF and Ni-Rh cells in a single cell. To verify the catalytic activity of the Ni-Rh cell for $CH_4$ conversion, we conducted the gas chromatography measurement at OCV conditions not to be affected by the electrochemical reaction, as shown in Supplementary Note 2 and Supplementary Figs. 9–12. The Ni-Rh cell exhibits the significantly higher $CH_4$ conversion, especially approaching thermodynamic equilibrium under S/C = 2, and lower

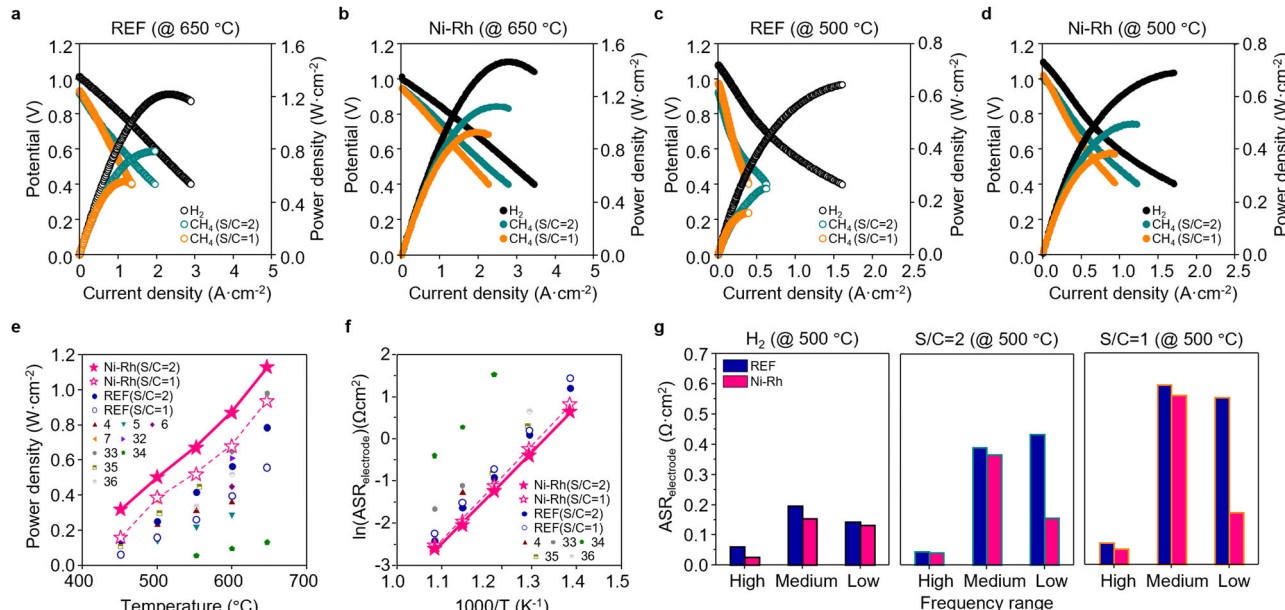

**Fig. 2 | Electrochemical performance evaluations of direct methane PCFCs.**
**a–d** Electrochemical performances of REF and Ni-Rh cells under different fuels ($H_2$, $CH_4$(S/C = 2) and $CH_4$(S/C = 1)) at 650 and 500 °C, where fuel conditions are 97% $H_2$ with 3% $H_2O$ for $H_2$ (100 sccm), 25% $CH_4$, 50% $H_2O$ and 25% Ar for S/C = 2 (32 sccm), and 25% $CH_4$, 25% $H_2O$ and 50% Ar for S/C = 1 (32 sccm), respectively. Air is fed into the cathode as an oxidant (100 sccm). Comparison of (**e**) the maximum power densities and (**f**) the area-specific polarization resistances with the previously reported PCFCs and SOFCs. **g** Area-specific polarization resistances according to different frequency ranges, high (>$10^3$ Hz), medium ($10$–$10^3$ Hz), and low (<10 Hz) frequencies, deconvoluted by DRT analysis under different fuel conditions ($H_2$, $CH_4$(S/C = 2), and $CH_4$(S/C = 1)).

activation energies (~26.6 kJ/mol) than those of the REF cell. Interestingly, the Ni-Rh cell shows a larger difference in $CH_4$ conversion than the REF cell under S/C = 2 (high $P(H_2O)$) rather than S/C = 1 (low $P(H_2O)$). It implies that the improvement in $CH_4$ activation with the Ni-Rh bimetallic catalyst is significantly associated with the water–catalyst interaction as well as the gas–catalyst interaction. In addition, the high surface coverage and maximized catalyst surface area properties with a small particle size enlarge the water–catalyst interaction, maximizing the electrochemical performance.

## Long-term stability and self-carbon cleaning mechanism
Long-term stability is the most challenging issue for the sustainable operation of direct methane PCFCs, mostly induced by carbon-coking which is a byproduct of methane steam reforming. Carbon-coking blocks the electrochemical and thermochemical reaction sites and rapidly degrades the electrochemical performance[38–41]. Figure 3 presents the long-term stabilities for REF and Ni-Rh cells under the S/C = 1 condition at 500 °C, where the carbon-coking is thermodynamically activated primarily by methane cracking ($CH_4 \rightarrow 2H_2 + C$) and the Boudouard reaction ($2CO \rightarrow CO_2 + C$) (Supplementary Fig. 10)[23,42,43]. As shown in Fig. 3(a), the REF cell shows a rapid decrease in the electrochemical performance with a degradation rate of 0.4%·h$^{-1}$ with a significant increase in $ASR_{electrode}$ over long-term operation (Supplementary Fig. 13a). In contrast, the Ni-Rh cell demonstrates a degradation rate of 0.02%·h$^{-1}$, which is ~20-fold lower than that of the REF cell, with almost unchanged $ASR_{electrode}$ (Supplementary Fig. 13b). Since we can eliminate the minor possibilities of degradation from the electrolyte and air electrode in the REF cell which has identical configurations with the Ni-Rh cell, low stability of the REF cell is primarily originated from the degradation of the fuel electrode performance. Furthermore, simultaneously decreased electrochemical performance and $CH_4$ conversion in the REF cell confirms that the fuel electrode performance of the REF cell substantially degrades over long-term operation. Postmortem analysis using energy dispersive spectroscopy (EDS) and Raman spectroscopy in Fig. 3b and Supplementary Figs. 14

and 15 clarify that the REF cell suffers from significant carbon-coking on the catalyst surface[12,44], as evidenced by the presence of carbon peaks (D band (disordered carbon; 1350 cm$^{-1}$) and G band (graphitic carbon; 1580 cm$^{-1}$))[4,45]. This carbon-coking deactivates the Ni surface, inhibiting the $CH_4$ activation and hydrogen oxidation reaction at the fuel electrode. On the other hand, the Ni-Rh cell shows no evidence of carbon-coking on the catalyst surface (Fig. 3b) and no carbon peaks in their spectra (Supplementary Fig. 14b), verifying the high tolerance against carbon-coking. In addition, although nanoparticles generally lose their active sites over long-term operation due to agglomeration, anchored Ni-Rh bimetallic catalysts at the BZCYYb surface show high structural stability without agglomeration, as shown in Supplementary Fig. 16. Therefore, this finding confirms that the Ni-Rh cell exhibits outstandingly robust chemical and structural stabilities under $CH_4$ operation without carbon-coking and agglomeration, preserving their active sites for gas reforming and electrochemical reactions.

Since the $CH_4$ operation of S/C = 1 at 500 °C is the thermodynamically favored regime for carbon-coking (Supplementary Fig. 10), high carbon-coking tolerance of the Ni-Rh cell implies the occurrence of self-carbon cleaning on the catalyst surface. The self-carbon cleaning process occurs through the following pathways (See Fig. 4a): 1) CO formation ($C^* + O^* \rightarrow CO^* + Ni^*$)[46], 2) CHO formation ($CH^* + O^* \rightarrow CHO^*$)[47], and 3) CHOH formation ($CH^* + OH^* \rightarrow CHOH^*$)[47] compared to the carbon-coking pathway, as shown in Supplementary Note 3. We conducted the in-situ DRIFTS measurements to observe the occurrence of the carbon cleaning process on the catalyst surface by the appearance of intermediate species of CO*, CHO*, and CHOH*[48–50], as shown in Fig. 4b, c. In the Ni-Rh cell, representative peaks associated with CO* (1664 cm$^{-1}$), CHO* (1420–1370 cm$^{-1}$), and CHOH* (1440–1400 cm$^{-1}$) emerge as the temperature increases to 500 °C. On the other hand, in the REF cell, only CHO* species appears. In addition to observation of the formyl group, the carbon cleaning process is substantially accompanied by the evolution of hydroxyl species (3750–3550 cm$^{-1}$)[51,52], as shown in Fig. 4d, e. The Ni-Rh cell exhibits a higher intensity of OH* than the REF cell. Moreover, the in-situ DRIFTS

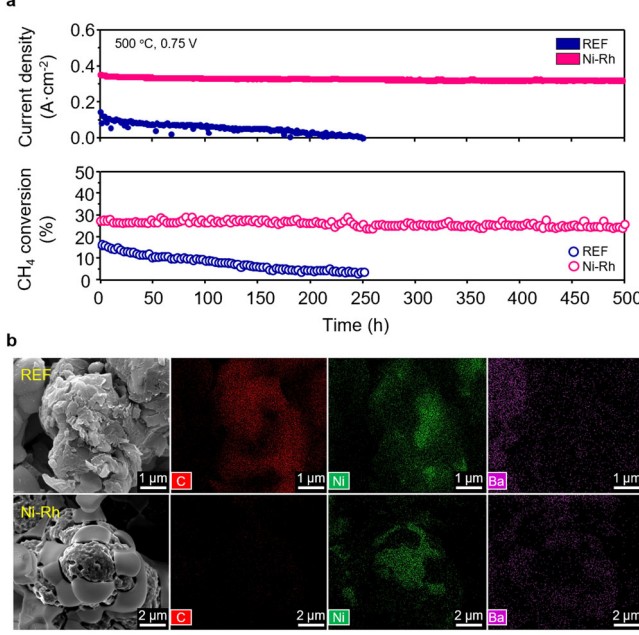

**Fig. 3 | Long-term stability of direct methane PCFC. a** Long-term stability evaluations of electrochemical performance and catalytic activity for 500 h under S/C = 1 conditions at 500 °C, where the direct methane PCFC operated with a fuel composition of 25% $CH_4$, 25% $H_2O$ and 50% Ar at the fuel electrode with a total flow rate of 100 sccm and air at the cathode as an oxidant (100 sccm) under the constant cell voltage of 0.75 V. **b** Postmortem EDS analysis after long-term operation.

results show the increase in the formyl group as the temperature increases, accompanied by a simultaneous increase in hydroxyl species. Therefore, we can conclude that the Ni-Rh cell has a self-carbon cleaning process by generating more formyl group from the evolution of hydroxyl species, indicating the higher carbon resistance than the REF cell. Detailed analysis of DRIFTS is explained in Supplementary Note 3 and Supplementary Figs. 17 and 18.

As well as DRIFTS studies, synchrotron-based in-situ XPS measurement in Fig. 5 further elucidates the correlations between self-carbon cleaning and evolution of $H_2O$ related-defects such as oxygen vacancies (Vo), hydroxyl groups (OH*), and oxidative species (O*)[4]. We measured the changes in the chemical natures of carbon (C-C $sp^3$ and C-Ni), oxygen defect species (OH*, Vo, and $O_O^\times$), and metallic catalyst (Ni and NiO) (Supplementary Note 4)[45]. In the REF cell, when $CH_4$ was fed, the relative area ratio of the C-C $sp^3$ and C-Ni spectra substantially increase by ~2.6 times and ~1.3 times, respectively, indicating carbon-coking during the reaction, as shown in Fig. 5a, b. On the other hand, in the Ni-Rh cell, the C-C $sp^3$ spectra remains almost unchanged; moreover, the C-Ni spectra disappears completely, clearly indicating the self-carbon cleaning process on the catalyst surface. As shown in Fig. 5c, d, the Ni-Rh cell forms more Vo and OH* than the REF cell. This phenomenon occurs because the Rh in the Ni-Rh bimetallic catalyst improves the dissociation of $H_2O^*$, thereby readily forming Vo and OH*[53–55]. In addition, the Ni-Rh bimetallic catalyst has a strong $H_2$ spillover effect, forming H* species on the BZCYYb surface[18–20]. These H* species react with OH* to form oxygen vacancies by a dehydration reaction (H* + OH* → $H_2O_{(g)}$ + Vo)[56] and react with lattice oxygen to form hydroxide (H* + $O_O^\times$ → OH*)[1] on the BZCYYb surface. The evolved Vo provides more sites for OH* formation (Vo + $O_O^\times$ + $H_2O_{(g)}$ → 2OH*)[1,2], thereby contributing to self-carbon cleaning[57]. Ni, with a lower electronegativity of 1.91 than Rh (2.29), attracts the O* species from $H_2O^*$ dissociation ($H_2O^*$ → OH* + H*) and subsequent OH* dissociation (OH* → O* + H*)[58,59]. On the other hand, in dry condition, degree of

carbon-coking and the formation of $H_2O$-related defects is not that different between the REF and Ni-Rh cells, as shown in Supplementary Fig. 19 and Supplementary Table 4. It reveals that the high $H_2O$ dissociation of Ni-Rh plays a critical role in the self-carbon cleaning process. Therefore, we conclude that the readily simple and cost-effective architecturing process for the Ni-Rh bimetallic catalyst at the fuel electrode is promising for the direct methane PCFCs. Furthermore, we believe that this approach is extensively applicable to other electrochemical devices that require the direct reforming of gases such as other hydrocarbon fuels, and ammonia.

## Discussion

In conclusion, we successfully demonstrated the high electrochemical performance and stability of direct methane PCFCs by modifying the fuel electrode with a self-assembled Ni-Rh bimetallic catalyst, which is readily fabricated by a one-step infiltration process with an extremely low amount of Rh. Changes in electrochemical performance according to different operating conditions and high resolution in-situ surface characterizations revealed that the Ni-Rh bimetallic catalyst exhibits strong water–catalyst interactions, simultaneously leading to high $CH_4$ conversion and self-carbon cleaning. Our results may broaden the utilization of direct alternative fuel PCFCs, such as other hydrocarbon fuels and ammonia based on their high performance and sustainability without engineering complexity. Furthermore, we expected that self-assembled bimetallic catalysts can be extensively applied to other electrochemical devices requiring enlarged catalytic active sites and robust structural stabilities.

## Methods

### Fabrication of fuel cells and methane steam reforming catalyst

An anode-supported single cell was fabricated with a configuration of NiO-BZCYYb/BZCYYb/PBSCF. NiO-BZCYYb anode powder was prepared using a mixture of homogeneous NiO powder (Kojundo Chemical), BZCYYb4411 ($BaZr_{0.4}Ce_{0.4}Y_{0.1}Yb_{0.1}O_{3-\delta}$) powder (Kceracell Co.), polymethyl methacrylate (PMMA) pore former (Grand Chemical & Material) at a weight ratio of 6:4:1 with a ball-milling process in ethanol with a 1 wt% dispersant (HypermerTM KD-6, Corda), 1.5 wt% polyvinyl butyral (Sigma Aldrich) binder, and 1.5 wt% dibutyl phthalate (Sigma Aldrich) plasticizer for 24 h. After ball-milling, the NiO-BZCYYb slurry was dried in a dry oven for a few hours. Dried NiO-BZCYYb powder was uniaxially pressed under 50 MPa and presintered at 1000 °C for 3 h to obtain mechanical strength and porosity. The anode functional layer (AFL) and BZCYYb electrolyte were deposited by slurry spin coating. AFL slurry was prepared with NiO and BZCYYb powders in a weight ratio of 6:4, mixed with a solvent (isopropyl alcohol), 2 wt% dispersant (BYK-2012, BYK), and 1.5 wt% binder (ethylcellulose). The electrolyte slurry was prepared using the same components and ratio as the AFL, except for using NiO powder. The spin-coated as-prepared single cell with AFL and electrolyte was cosintered at 1500 °C for 5 h to densify the electrolyte and grow the grain. For the cathode, the $PrBa_{0.5}Sr_{0.5}Co_{1.5}Fe_{0.5}O_{5+\delta}$ (PBSCF) powder (Kceracell Co.) was mixed with the binder ink (VEH, Fuel Cell Materials) at a weight ratio of 1:1 to fabricate porous cathode structures, and it was screen-printed onto the BZCYYb electrolyte to obtain a thickness of ~17 μm with effective area of 0.16 $cm^2$ (Anode size of 0.785 $cm^2$). After screen printing, the single cell with the cathode layer was sintered at 950 °C for 4 h.

Ni-Rh bimetallic alloy catalysts in the fuel electrode substrate were fabricated by one-step infiltration. The precursor solution for infiltration was 0.05 molarity (mol/L), which was mixed with Rh nitrate ($Rh(NO_3)_3 \cdot xH_2O$, Sigma–Aldrich) and ethanol solvent without any materials such as dispersant and binder. In the one-step infiltration method, the precursor solution was applied once to the porous surface of the fuel electrode substrate (NiO-BZCYYb), followed by calcination at approximately 300 °C for 1 h. Through one-

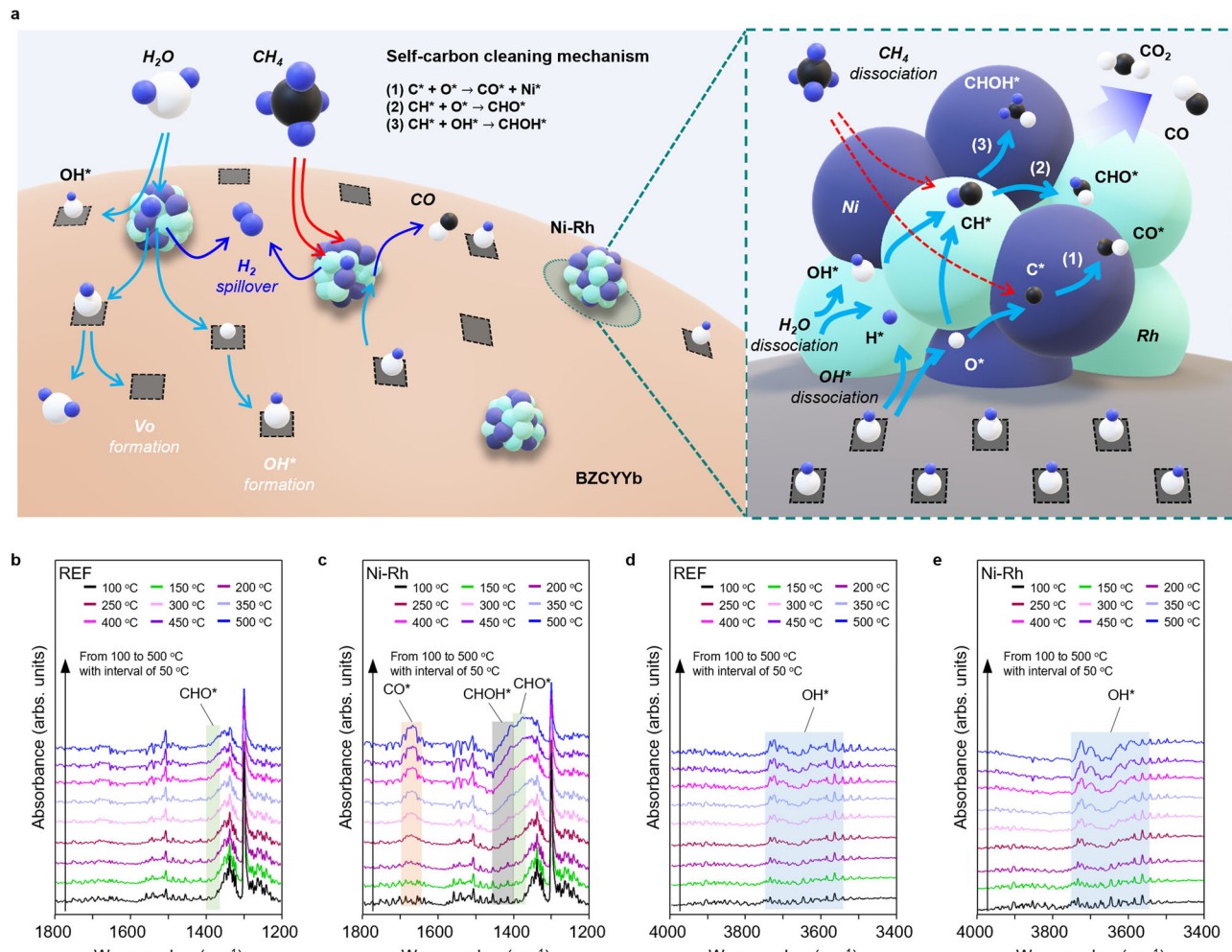

**Fig. 4 | Self-carbon cleaning mechanism on Ni-Rh bimetallic catalyst.**
**a** Schematic diagram of self-carbon cleaning on the Ni-Rh bimetallic catalyst. In-situ DRIFTS analysis at different wavenumber range of 1800–1200 cm⁻¹ (**b** REF and **c** Ni-Rh cells) and 4000–3400 cm⁻¹ (**d** REF and **e** Ni-Rh cells), respectively, during steam reforming of methane (3% $CH_4$, 3% $H_2O$ and 94% Ar for S/C = 1, 20 sccm) in the temperature range of 100–500 °C.

step infiltration, we used an extremely low amount of Rh for a cell relative to Ni (~224 mg/cm² for Ni and ~0.14 mg/cm² for Rh); thus, the price of Rh was ~28.5-fold lower than that of Ni ($2.3 \times 10^{-4}$ \$/cm² for Rh (\$ 765.6/lb, Daily Metal Prices) and $6.6 \times 10^{-3}$ \$/cm² for Ni (\$ 13.3/lb, Daily Metal Prices)).

### Characterization of microstructure and postmortem analysis

The morphology and microstructure of the fuel electrode substrate were observed by field emission scanning electron microscopy (FE-SEM; Inspect F, FEI). SEM-energy dispersive X-ray spectroscopy (EDS) was performed to investigate the carbon formed on the fuel electrode surface, and surface coverage was calculated by processing software (ImageJ). The particle size and chemical composition of the catalyst formed on the surface of the fuel electrode were examined by high-resolution transmission electron microscopy (HR-TEM; NEOARM JEM-ARM 200 F, JEOL) and energy dispersive X-ray spectroscopy (EDS), in which samples were prepared by a focused ion beam (FIB; crossbeam 540, ZEISS). After the long-term stability test, the XRD patterns of the fuel electrode substrate were recorded by using a D8 Advance (Bruker) using Cu Kα radiation to characterize the crystal structure. The crystal structure of the fuel electrode substrate was scanned with a step size of 0.02°/s in the 2θ range = 25 to 80°. To obtain carbon formation information on the fuel electrode substrate after the long-term stability test, Raman spectroscopy with LabRam Aramis (Horiba Jobin Yvon)

was performed in the range of 1000 to 2000 cm⁻¹ with a yag laser (λ = 532 nm).

### Electrochemical evaluation of fuel cells

Electrochemical impedance spectra (EIS) measurement in the frequency ranges from 1 MHz to 0.1 Hz at open circuit voltage (OCV), and current-voltage (I-V) measurements were performed using IviumStat.h (Ivium Technologies). The electrochemical evaluation at operating conditions was carried out at 650–500 °C under various fuels (i.e., $H_2$ and $CH_4$ with a steam to carbon ratio (S/C) of 2 and 1) and air (at a flow rate of 100 sccm). $H_2$ fuel operation was 100 sccm of $H_2$ (with 3% $H_2O$), and $CH_4$ fuel operation was 32 sccm of mixed gases (S/C = 2: 25% $CH_4$, 50% $H_2O$, and 25% Ar; S/C = 1: 25% $CH_4$, 25% $H_2O$, and 50% Ar). Dry gases (Air, $H_2$, $CH_4$, and Ar) were controlled by the mass flow controller (MFC, Bronkhorst). Steam was controlled using the saturation temperature in the humidifier to control the steam concentration in the feed gas.

### In-situ DRIFTS

In-situ diffuse reflectance infrared Fourier transform spectroscopy (DRIFTS) experiments were performed using DiffusIR™ (PIKE Technologies) to examine the adsorbed species generated during the steam reforming of methane reaction. Spectra were obtained with 64 scans with a resolution of 4 cm⁻¹ in the range of 4000–650 cm⁻¹ through an

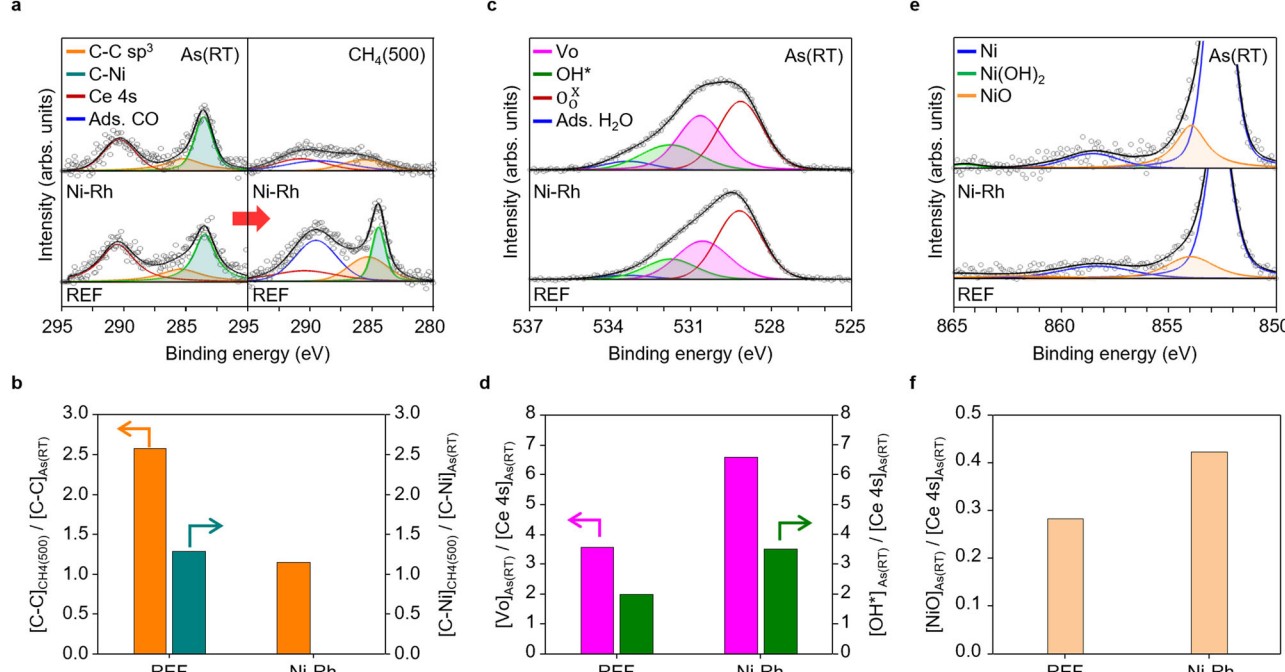

**Fig. 5 | Effects of defect species on self-carbon cleaning process.** Synchrotron-based in-situ XPS measurement of (**a**) C $1s$ photoelectron spectra at the initial state (room temperature) and CH$_4$ feeding condition (500 °C). **b** Relative ratio of carbon amount between that at the initial state and at the operation state to quantify the carbon-coking ([C-C]$_{CH4(500)}$/[C-C]$_{As(RT)}$ and [C-Ni]$_{CH4(500)}$/[C-Ni]$_{As(RT)}$) in REF and Ni-Rh cells. **c** O $1s$ photoelectron spectra at the initial state. **d** Concentration of oxygen vacancy (Vo) and hydroxyl group (OH*), **e** Ni $2p$ photoelectron spectra at the initial state, and (**f**) concentration of NiO.

MCT detector cooled by a liquid N$_2$. The samples (REF and Ni-Rh) have a porous structure that mimics a fuel electrode and were loaded into a porous ceramic alumina cup which was placed in a heatable holder. Before confirming the self-carbon cleaning, the samples in the DRIFTS chamber were reduced with mixed H$_2$ gas (10% H$_2$, 90% Ar, 20 sccm) at 650 °C for 1 h, then cooled with Ar gas (100% Ar, 20 sccm) to 100 °C, and the background was recorded with the reduced samples. After the pretreatment process, the in-situ DRIFTS experiment measured from 100 °C to 500 °C during injection of the mixed gas (3% CH$_4$, 3% H$_2$O, 94% Ar, 20 sccm), and each temperature was maintained for 30 min to reach steady state conditions.

### Synchrotron-based in-situ HR−XPS

Two types (wet and dry) of samples were prepared to confirm the self-carbon cleaning at the fuel electrode surface during the methane reforming reaction when exposed to H$_2$O. Samples (REF and Ni-Rh) were exposed to 3% wet Ar gas at room temperature for one day after reduction at 650 °C and were designated wet samples. The samples (REF, Ni-Rh) that were not exposed to wet conditions were designated dry samples.

In-situ high-resolution X-ray photoelectron spectroscopy (HR-XPS) was measured by the 10A2 beamline in the Pohang Accelerator Laboratory (PAL). During in-situ HR-XPS analysis, the fuel electrode substrates were fixed to a molybdenum holder and placed in an ultrahigh vacuum (UHV) chamber maintained at a base pressure of $5 \times 10^{-10}$ torr. The binding energies and spectral resolutions were calibrated by recording the Au 4 f photoelectron peak as a reference (4f$^{7/2}$, BE = 84.0 eV). Photon energy was used at the same excitation energy (960 eV) for measurement reproducibility for all samples.

To obtain a clean surface, the samples were subjected to Ar sputtering (1500 L = $5 \times 10^{-6}$ Torr for 300 s at 1 keV), and the surfaces of the fuel electrode substrates (As(RT)) were measured before the reaction. After measurements, the surface states of the fuel electrode substrates (CH$_4$(500)) were measured by exposing the samples to a reaction gas (25% CH$_4$/Ar balance) for 3000 L ($5 \times 10^{-6}$ Torr for 600 s) at 500 °C.

## Data availability

All data generated in this study are provided in the manuscript and Supplementary Information file. Source data file is provided in this paper. Source data are provided with this paper.

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

## Acknowledgements

J.H. acknowledges the support from the Ministry of Trade, Industry and Energy (MOTIE), South Korea and Korea Institute for Advancement of Technology (KIAT), South Korea through the International Cooperative R&D program (P0021202). J.H., W.L., and M.C. appreciate the support by the National Research Foundation of Korea (NRF) grant funded by the Korea government (MSIT) (2023M3J1A1091543, 2022R1A2C3012372, 2022R1A4A1031182, 2021K1A3A1A20002574, and 2021R1C1C2006657). M.C. thanks the support from the funds of the Open R&D program of Korea Electric Power Corporation (R23XO03).

## Author contributions

K.H., and M.C.: Conceptualization of the work and experimental design and investigation. Y.B., J.M.: GC measurement and data analysis. J.L., D.K., S.B.: Cell fabrication and cell characterizations using SEM and TEM. H.K.L.: Synchrotron-based in-situ XPS measurement and data analysis. W.L., J.H.: Supervision and management of the research project. All authors discussed the results and commented on the manuscript.

## Competing interests

The authors declare no competing interests.
