## [Peer Review File · Nature Communications]

Direct methane protonic ceramic fuel cells with self-assembled Ni-Rh bimetallic catalystREVIEWER COMMENTS

Reviewer #1 (Remarks to the Author):

The article reports on the enhanced catalytic activity of Rh nanoparticles when infiltrated in fuel electrodes on protonic ceramic fuel cells to act together with the Ni bulk and exsolved Ni particles there. The Rh is shown to improve steam reforming and reduce coking in the case where CH₄+steam is used as fuel, and also therefore to improve stability during operation with such fuels.

I find the work technologically and scientifically well performed, interpreted, and reported. The SI is extensive and solid. Some of the results in the main paper are however too detailed and could better be moved to the SI.

The main problem is that the paper in my opinion has severe flaws in its presentation, i.e., the text and argumentation and logics. The English is generally good (except that coking should be coking), so the problem lies more elsewhere. In the following I point out some examples.

- the use of extremes or other meaningless adjectives in phrasing is inappropriate. It is partly deliberate, partly inadvertent. The title's "exceptional performance" is a first example. It is better but hardly exceptional. Moderate it. Line 69: "extremely low" must similarly be moderated.

- The Abstract is bad. It starts with a abbreviation PCFCs that has not been defined. It follows up with "Direct methane PCFCs are promising electrochemical devices that address the technical and economic challenges associated with using pure hydrogen, such as the high cost of green production, transportation, and long-term storage." I think that the PCFCs address some of these, but not all. The sentence is too imprecise. In any case, the Abstract need not be a full introduction, but go to the point of the work. "coking" must be changed to "coking" throughout. The sentence "The Ni-Rh bimetallic catalyst shows remarkably high catalytic activity with an exceptional performance of ~0.50 W/cm² at 500 °C." mixes its role as catalyst with the performance of the cell. The final sentence that the Ni-Rh catalyst "...initiates a self-carbon cleaning process..." is too obscure in my opinion. All in all, the Abstract needs a rehab.

- In the Introduction, the authors state that "there is still a lack of required technologies for green hydrogen production, storage, and transportation." This is nonsense - H₂ has been produced large scale by green electrolysis for >100 years, stored, and transported. We would like, however, to improve the efficiency. But that's a different matter. The authors jump too easily to cheap statements and conclusions.

- The sentence in lines 50-51 is awkward and needs to be reorganised.

- Line 60: The metals are not novel, they are noble.

- Lines 79-80. Something wrong with the phrasing. Plus, here - and elsewhere in the manuscript - the authors seem to state that carbon dioxide is a fuel...

- In Results and Discussion, Main 1. Fig. 1 a is hard to understand, and both this and b-e need more and better caption text.

- In lines 94-97 and several places later on, the authors use a kind of defect chemical reactions to explain what they think happens. I think it is unfortunate that the equations don't conform to the rules of defect chemistry and become a quite messy contribution of the paper. Since exsolution etc. deal with more than one phase, it is not easy to know what formalism to use, but I think the authors have not succeeded in their attempt. The problem repeats itself in numerous additional occasions later on...

- Lines 104 and 115 have some misprints....

- Conclusions: Again, CO₂ seems to be mentioned as a fuel....

- Methods: Line 292 uses the word "monitored" but I think "controlled" would be more correct. And I think Eqs. 1-3 are too trivial to deserve space in the publication. They could well have been moved to SI if needed at all.

Reviewer #2 (Remarks to the Author):

The direct methane PCFCs are worthy to study and the new Ni-Rh alloy nanoparticle catalysts achieved improved performance compared to the conventional Ni catalyst. However, I did have the following comments that need to be addressed before I can make a final suggestion.

- 1) Some experimental details should be given. For example, the amounts of ethanol, dispersant, binder, and plasticizer.
- 2) The most important information on cell effective area was not clearly given, which affects the scalability.
- 3) The BZCYYb electrolyte might have oxygen ion conductivity, which allowed the oxygen ion from the cathode side to fuel cells to burn carbon when the low S/C ratio was used. How was this effect excluded from the explanation?
- 4) The methane conversion is still low. The anode exhaust still has more methane, and carbon monoxide. The CO₂ purity is low, which still needs further treatment. How can methane conversion be increased? The equilibrium shift to the right side by removing hydrogen should be considered. The conventional thermodynamic equilibrium should not be the boundary. Have the authors tried to lower the space velocity to improve the methane conversion?
- 5) The calculation detail for the equilibrium methane conversion should be given since some hydrogen is transported to the cathode side and formed water. The hydrogen was removed from the steam reforming system, the conventional equilibrium was shifted to the right side.
- 6) It is better to get the ASR comparison for the fuel electrode only.

Reviewer #3 (Remarks to the Author):

In terms of the power density and durability, the results demonstrated in this work are good. However, there are a lot of experiments should be performed to clarify some results and strengthen the conclusions made in this work. The following concerns should be addressed before making the decision.

1. It is no clear if the Ni-Rh are bimetallic alloy. The TEM images are not clear and

insufficient. Additional experiments should be performed to further confirm it.

2. No TEM images of the anode prior to reduction were provided. It is unclear if the Rh is exsolved from the lattice.

3. Figure 2a-2b, why the Ni-Rh PCFC performance under hydrogen was also significantly improved? Typically, the anode does not greatly affect the PCFC performance. Although figure 2g provides the ASRp, it is suggested to perform additional experiments and in-depth analysis to prove the performance improvement is due to the anode.

4. Figure 2d, the IV curve under methane ($s/c=2$) is weird. It is suggested to retest it.

5. The anode reforming activity is not evaluated in a packed bed reactor and compared with the reference anode. Otherwise, it is not clear to me if the performance improvement is due to the anode or something else. Therefore, anode reforming activity should be also evaluated in a packed bed reactor.

6. There is no evidence why the REF PCFC is unstable under methane. Is that due to coking or something else? More evidence should be provided.

7. The proposed mechanism in Figure 4 looks pretty. However, there is a lack of evidence to support it. It is suggested to perform additional experiments or conduct computational modeling to support this mechanism.

8. Additionally, there is lack of details provided in the paper/figures, making it hard to understand the PCFC testing conditions. For example, Figure 3, what's the current density? There are a lot of similar issues.

Respond to Revision

Reviewer #1 (Remarks to the Author): The article reports on the enhanced catalytic activity of Rh nanoparticles when infiltrated in fuel electrodes on protonic ceramic fuel cells to act together with the Ni bulk and exsolved Ni particles there. The Rh is shown to improve steam reforming and reduce coking in the case where CH₄+steam is used as fuel, and also therefore to improve stability during operation with such fuels.

I find the work technologically and scientifically well performed, interpreted, and reported. The SI is extensive and solid. Some of the results in the main paper are however too detailed and could better be moved to the SI.

The main problem is that the paper in my opinion has severe flaws in its presentation, i.e., the text and argumentation and logics. The English is generally good (except that cocking should be coking), so the problem lies more elsewhere. In the following I point out some examples.

1. the use of extremes or other meaningless adjectives in phrasing is inappropriate. It is partly deliberate, partly inadvertent. The title's "exceptional performance" is a first example. It is better but hardly exceptional. Moderate it. Line 69: "extremely low" must similarly be moderated.

The manuscript has been extensively revised to correct all typos such as 'cocking' to 'coking' and to change exaggerating words such as 'exceptional' and 'extremely' to more moderate adjectives. We highlighted the changes in the manuscript as follows:

R1

- Line 1, Title

[Before]

Self-assembled Ni-Rh bimetallic catalyst for the **exceptional** performance and stability of direct methane protonic ceramic fuel cells

[After]

Self-assembled Ni-Rh bimetallic catalyst for the **high** performance and stability of direct methane protonic ceramic fuel cells

- Line 30-33

[Before]

The Ni-Rh bimetallic catalyst shows remarkably high catalytic activity with an **exceptional** performance of ~ 0.50 W/cm² at 500 °C.

[After]

Herein, we introduce a self-assembled Ni-Rh bimetallic catalyst that exhibits a significantly high CH₄ conversion and carbon-coking tolerance. It enables direct methane PCFC to operate with a **high** peak power density of ~ 0.50 W/cm² at 500 °C, surpassing all other previously reported values from direct methane PCFC and even solid oxide fuel cells.

- Line 65-66

[Before]

In this study, we report a direct methane PCFC with **exceptional** performance and stability with a self-assembled Ni-Rh bimetallic catalyst.

[After]

In this study, we report a direct methane PCFC with **high** performance and stability with a self-assembled Ni-Rh bimetallic catalyst.

- Line 68-70

[Before]

PCFC with a self-assembled Ni-Rh bimetallic catalyst exhibits an **exceptionally** high performance of 1.13 W/cm^2 at $650 \text{ }^\circ\text{C}$ and 0.50 W/cm^2 at $500 \text{ }^\circ\text{C}$ under direct methane fuel conditions, surpassing other previously reported direct methane PCFCs and solid oxide fuel cells (SOFCs).

[After]

PCFC with a self-assembled Ni-Rh bimetallic catalyst exhibits a **significantly** high performance of 1.13 W/cm^2 at $650 \text{ }^\circ\text{C}$ and 0.50 W/cm^2 at $500 \text{ }^\circ\text{C}$ under direct methane fuel conditions, surpassing other previously reported direct methane PCFCs and solid oxide fuel cells (SOFCs).

- Line 137-139

[Before]

These **exceptional** MPDs of the Ni-Rh cell are primarily attributed to the lowest area-specific polarization resistance ($ASR_{\text{electrode}}$), corresponding to the electrode resistance; the values are exceptional relative to other reported values, as shown in Fig. 2(f).

[After]

These **higher** MPDs of the Ni-Rh cell are primarily attributed to the lowest area-specific polarization resistance ($ASR_{\text{electrode}}$), corresponding to the electrode resistance as shown in Fig. 2(f).

2. The Abstract is bad. It starts with an abbreviation PCFCs that has not been defined. It follows up with "Direct methane PCFCs are promising electrochemical devices that address the technical and economic challenges associated with using pure hydrogen, such as the high cost of green production, transportation, and long-term storage." I think that the PCFCs address some of these, but not all. The sentence is too imprecise. In any case, the Abstract need not be a full introduction, but go to the point of the work. "cocking" must be changed to "coking" throughout. The sentence "The Ni-Rh bimetallic catalyst shows remarkably high catalytic activity with an exceptional performance of $\sim 0.50 \text{ W/cm}^2$ at $500 \text{ }^\circ\text{C}$." mixes its role as catalyst with the performance of the cell. The final sentence that the Ni-Rh catalyst "...initiates a self-carbon cleaning process..." is too obscure in my opinion. All in all, the Abstract needs a rehab.

We modified the abstracts as follows.

[Abstract]

Direct methane protonic ceramic fuel cells (PCFCs) are promising electrochemical devices that address the technical and economic challenges of conventional ceramic fuel cells. However, Ni, a catalyst of PCFC exhibits sluggish reaction kinetics for CH_4 conversion and a low tolerance against carbon-coking, limiting its wider applications. Herein, we introduce a self-assembled Ni-Rh bimetallic catalyst that exhibits a significantly high CH_4 conversion and carbon-coking tolerance. It enables direct methane PCFC to operate with a high maximum power density of $\sim 0.50 \text{ W/cm}^2$ at $500 \text{ }^\circ\text{C}$, surpassing all other previously reported values from direct methane PCFC and even solid oxide fuel cells. Moreover, it allows stable operation with a degradation rate of $0.02\%/h$ at $500 \text{ }^\circ\text{C}$ over 500 h, which is ~ 20 -fold lower than that of

conventional PCFC (0.4%/h). High-resolution in-situ surface characterization techniques reveal that high-water interaction on the Ni-Rh surface facilitates the carbon cleaning process, enabling sustainable long-term operation.

3. In the Introduction, the authors state that "there is still a lack of required technologies for green hydrogen production, storage, and transportation." This is nonsense - H₂ has been produced large scale by green electrolysis for >100 years, stored, and transported. We would like, however, to improve the efficiency. But that's a different matter. The authors jump too easily to cheap statements and conclusions.

We corrected the introduction in a way that emphasizes the importance of direct methane protonic ceramic fuel cells without unclear speculations, ambiguous arguments, or exaggerating expressions.

4. The sentence in lines 50-51 is awkward and needs to be reorganized.

We reorganized the sentences as follows.

[Before]

Nevertheless, there are several limitations to utilizing direct methane PCFCs. At low temperatures (< 600 °C), the catalytic activity of Ni, which is the monometallic catalyst at the fuel electrode of PCFC, for CH₄ activation decreases rapidly due to sluggish reaction kinetics, creating an insufficient hydrogen supply¹¹

[After]

However, several challenges must be addressed for the direct methane PCFCs. First, conventional catalysts such as nickel (Ni) exhibit substantially reduced CH₄ reforming activity as the operating temperature is lowered, which induces insufficient hydrogen supply¹¹.

5. Line 60: The metals are not novel, they are noble.

The manuscript has been extensively revised to correct all typos such as 'novel' to 'noble'.

6. Lines 79-80. Something wrong with the phrasing. Plus, here - and elsewhere in the manuscript - the authors seem to state that carbon dioxide is a fuel...

We rephrased the sentence as follows and deleted the carbon dioxide in the sentences.

[Before]

We believe that the bimetallic catalyst self-assembled by a readily simple and cost-effective process, and it is extensively applicable in electrochemical cells with various gases, such as ammonia, carbon dioxide, and hydrocarbon fuels.

[After]

Our approach, a self-assembled bimetallic catalyst, is readily simple and cost-effective, enabling the extensive application to other electrochemical cells that requires the reforming of other various gases such as hydrocarbon fuels and ammonia.

7. In Results and Discussion, Main 1. Fig. 1 a is hard to understand, and both this and b-e need more and better caption text.

We modified the figure and added more details in the caption as follows to improve the readability.

[Before]

Fig. 1. Self-assembly process of the Ni-Rh bimetallic catalyst. (a) Schematics of the fuel electrode formation process for the REF cell (exsolved Ni particles) and Ni-Rh cell (self-assembly between infiltrated Rh and exsolved Ni particles). High magnification SEM images of the fuel electrode morphology for the (b) REF cell with exsolved Ni and (c) Ni-Rh cell with a self-assembled Ni-Rh bimetallic catalyst. Structure and chemical composition of the catalyst on the fuel electrode according to TEM and EDS mapping for the (d) REF and (e) Ni-Rh cells.

[After]

Fig. 1. Self-assembly process of the Ni-Rh bimetallic catalyst. (a) Schematics of the self-assembly process between infiltrated Rh and exsolved Ni particles. High magnification SEM images of the fuel electrode morphology of Ni-Rh cell (b) before reduction and (c) after reduction, respectively. Structure and chemical composition of the catalyst on the fuel electrode by TEM and EDS mapping with lattice spacing images for the Ni-Rh cell (d) before reduction and (e) after reduction, respectively.

8. In lines 94-97 and several places later on, the authors use a kind of defect chemical reactions to explain what they think happens. I think it is unfortunate that the equations don't conform to the rules of defect chemistry and become a quite messy contribution of the paper. Since exsolution etc. deal with more than one phase, it is not easy to know what formalism to use, but I think the authors have not succeeded in their attempt. The problem repeats itself in numerous additional occasions later on...

As the reviewer pointed out, the suggested defect chemistry formulas have rather ambiguous points due to the lack of experimental evidences to support this argument. Reflecting reviewer's comment, we corrected this phrase more concisely and edited it clearer to improve the readability.

[Before]

During sintering, Ba is evaporated from the BZCYYb lattice, creating an A-site vacancy ($ABO_{3-\delta} \rightarrow A_{1-\alpha}BO_{3-\delta} + \alpha(V''_{A-site} + V_o'') + \alpha AO_{(g)}$)²². The formation of A-site vacancies facilitates Ni diffusion into the perovskite oxide lattice as an interstitial defect ($NiO + V''_{A-site} + V_o'' \rightarrow (Ni_{i,V_{A-site}})^{\times} + O_o^{\times}$)^{23,24}. Diffused Ni is exsolved from the lattice during reduction ($(Ni_{i,V_{A-site}})^{\times} + O_o^{\times} \rightarrow V''_{A-site} + V_o'' + Ni_{ex-solved}$)^{25,26}. Thus, the fuel electrode with Ni-diffused BZCYYb serves as a platform for the subsequent self-assembly of Ni-Rh bimetallic catalysts. As shown in Supplementary Fig. 2, exsolved Ni particles are more evident as the sintering temperature increases due to facilitated Ba evaporation and Ni diffusion. We cosintered the fuel electrode and electrolyte at 1500 °C to achieve the largest grain size and sufficient densification of the electrolyte and to form exsolved

Ni particles. For the self-assembly of the bimetallic catalyst, we decorated the surface of Ni-diffused BZCYYb with Rh nanoparticles through a one-step infiltration process. Since Rh is highly miscible with Ni, the infiltrated Rh spontaneously mixes with subsequently exsolved Ni during H₂ reduction ($\text{Ni}_{i,V_{A\text{-site}}}^{\times} + \text{O}_{\text{O}}^{\times} + \text{Rh}_{\text{surface}} \rightarrow \text{V}_{A\text{-site}}^{\prime\prime} + \text{V}_{\text{O}}^{\prime\prime} + (\text{NiRh})_{\text{ex-solved}}$), resulting in a self-assembled Ni-Rh bimetallic catalyst in the fuel electrode.

[After]

To architecture the fuel electrode with a self-assembled Ni-Rh bimetallic catalyst, we combine the exsolution and one-step infiltration processes on the Ni/BZCYYb anode-support single cell configuration. We first deliberately designed a Ni-diffused BZCYYb fuel electrode using the interdiffusion mechanism of Ni (See Supplementary Fig. 1) and utilized it as a platform for the subsequent self-assembly between exsolved Ni particles and infiltrated Rh particles to form bimetallic catalysts. As shown in Fig. 1(a), we decorated the surface of Ni-diffused BZCYYb with Rh nanoparticles through the one-step infiltration process. Since Rh is highly miscible with Ni, the infiltrated Rh particles are autonomously mixed with subsequently exsolved Ni during H₂ reduction, resulting in a Ni-Rh bimetallic catalyst. Within the applicable temperature range in this study, we sintered the fuel electrode at 1500 °C, which demonstrates the largest grain size, to reduce the ohmic resistance and to facilitate the Ni-Rh bimetallic alloy formation through the large number of Ni exsolution (See Supplementary Fig. 2). We denoted the cell without Rh decoration and with Rh decoration as REF and Ni-Rh cell, respectively.

9. Lines 104 and 115 have some misprints....

We corrected the misprints.

10. Conclusions: Again, CO₂ seems to be mentioned as a fuel....

We corrected the sentence to remove CO₂.

11. Methods: Line 292 uses the word "monitored" but I think "controlled" would be more correct. And I think Eqs. 1-3 are too trivial to deserve space in the publication. They could well have been moved to SI if needed at all.

We corrected the word and moved the Eqs.1-3 to the SI as the reviewer suggested.

Reviewer #2 (Remarks to the Author): The direct methane PCFCs are worthy to study and the new Ni-Rh alloy nanoparticle catalysts achieved improved performance compared to the conventional Ni catalyst. However, I did have the following comments that need to be addressed before I can make a final suggestion.

1. Some experimental details should be given. For example, the amounts of ethanol, dispersant, binder, and plasticizer.

We provided the additional experimental details and highlighted in the manuscript.

2. The most important information on cell effective area was not clearly given, which affects the scalability.

The effective area of the fabricated cell is 0.16 cm^2 . We added information in the "Methods" section of the manuscript Line 269.

3. The BZCYYb electrolyte might have oxygen ion conductivity, which allowed the oxygen ion from the cathode side to fuel cells to burn carbon when the low S/C ratio was used. How was this effect excluded from the explanation?

As the reviewer pointed out, some proton conductors have oxygen ionic conductivity through the oxygen vacancies inside their lattice. Thus, discovering this effect on the CH_4

conversion might be an interesting point. However, according to the recent study from Duan et al., (Nature Energy 4.3 (2019): 230-240), the oxygen ion transference number of the positive electrode (cathode side) of BZCYYb is almost zero especially at 500 °C (Transference number (oxygen ion conductivity / total ion conductivity) = 0.02 at 500 °C). It can be ascribed that BZCYYb is a nearly pure proton conductor at the low temperature region. Therefore, since our operating temperature for long-term stability test is 500 °C, the effect of removing carbon by conducted oxygen ion from cathode would be very minor.

4. The methane conversion is still low. The anode exhaust still has more methane, and carbon monoxide. The CO₂ purity is low, which still needs further treatment. How can methane conversion be increased? The equilibrium shift to the right side by removing hydrogen should be considered. The conventional thermodynamic equilibrium should not be the boundary. Have the authors tried to lower the space velocity to improve the methane conversion.

Methane conversion can be determined by experimental conditions such as space velocity, S/C ratio, and temperature. The effect of temperature and S/C ratio is included in this paper, and that of space velocity was confirmed in our previous study (Hong et al., Journal of Materials Chemistry A 9.10 (2021): 6139-6151.). As the reviewer pointed out, increasing the residence time by lowering the space velocity can enhance the methane conversion by providing sufficient reaction time between methane and the catalyst. In our previous study, we have evaluated the steam reforming of methane supported by Ni-Rh/BZCYYb in the powder-based packed bed reactor under various temperatures (Figure R1(a)), S/C ratios

(Figure R1(b)), and space velocities (Figure R1(c)). As shown in Figure R1(c), methane conversion indeed increases as the space velocity is lowered. This is because the lower space velocity enables reactants to have more sufficient time to proceed reforming reactions. If the space velocity is further reduced, the overall process is no longer limited by chemical reaction rates, and rather governed by mass transport within a reactor. Hence, it is difficult to evaluate the intrinsic catalytic activity of the catalyst at a lower space velocity.

The primary investigation of this manuscript is to demonstrate the intrinsic catalytic activity improvement by forming Ni-Rh bimetallic alloy at the fuel electrode of a single cell and to elucidate its impact on electrochemical performance of direct methane PCFC. Therefore, we conducted the experiment at a high space velocity to confirm the intrinsic catalytic activity more clearly.

The response to the change in thermodynamic equilibrium by removed hydrogen through electrochemical reaction is addressed in comment #5.

Figure R1. Evaluation of CH₄ conversion under packed bed reactor as a function of (a) temperature, (b) S/C ratio, and (c) space velocity. (Hong et al., Journal of Materials Chemistry A 9.10 (2021): 6139-6151.)

5. The calculation detail for the equilibrium methane conversion should be given since some hydrogen is transported to the cathode side and formed water. The hydrogen was removed from the steam reforming system, the conventional equilibrium was shifted to the right side.

As the reviewer mentioned, when the electrical current is applied, hydrogen is consumed electrochemically in the fuel electrode, and heat released from exothermic reactions raises the local temperature. This would change the thermodynamic equilibrium following the changes in internal temperature and gas partial pressure. However, in this study (Supplementary Figure 9 and 11), the CH₄ conversion test was conducted in the OCV state in order to evaluate the intrinsic thermochemical reactions in the fuel electrode not affected by the electrochemical reaction. It was intended to elucidate the pure intrinsic catalytic activity of the Ni-Rh bimetallic alloy catalyst developed in the fuel electrode for steam reforming of methane. Therefore, we calculated the equilibrium considering the composition of the injected fuel composition (steam to carbon ratio of 2 and 1) and the operating temperatures (650-450 °C) under the OCV state. The detailed experimental conditions were added to the Supporting information and highlighted.

6. It is better to get the ASR comparison for the fuel electrode only.

First of all, since other components such as electrolyte and cathode are all identical between REF and Ni-Rh cell except the Rh infiltration at the fuel electrode, we believe that the significant change in low frequency resistance in Figure 2(g) is largely attributed to the fuel electrode performance of Ni and Ni-Rh fuel electrode.

As the reviewer commented, comparing the fuel electrode ASR would be better to compare the fuel electrode performance. However, distinguishing the sole fuel electrode resistance from the entire ASR in a single cell is challenging. Thus, we tried to compare the fuel electrode performance between REF and Ni-Rh cell by deconvoluting the high ($> 10^3$ Hz), medium ($10-10^3$ Hz), and low (< 10 Hz) frequency resistances corresponding to the charge transfer at the triple phase boundary (TPB) of the fuel and air electrodes, the gas adsorption process and the overall surface reactions at the electrodes, and the gas diffusion and fuel reforming in the fuel electrode, respectively (Line 161-178 in the manuscript).

Moreover, for in-depth analysis as the reviewer suggested, we conducted additional symmetric cell analysis of the REF and Ni-Rh fuel electrode under H_2 and CH_4/H_2O environments to further verify the effect of Ni-Rh on the decrease in ASR_p , as shown in Figure R2. Under the H_2 environment, the Ni-Rh cell shows the slight decrease in ASR_p ($0.108 \text{ ohm}\cdot\text{cm}^{-2}$ at $500 \text{ }^\circ\text{C}$) compared to that of the REF cell ($0.140 \text{ ohm}\cdot\text{cm}^{-2}$ at $500 \text{ }^\circ\text{C}$). Under the CH_4/H_2O environment, ASR_p of the Ni-Rh cell shows a ~ 2 -fold smaller value ($0.249 \text{ ohm}\cdot\text{cm}^{-2}$ at $500 \text{ }^\circ\text{C}$) than that of the REF cell ($0.565 \text{ ohm}\cdot\text{cm}^{-2}$ at $500 \text{ }^\circ\text{C}$), which is more significant difference compared to that under H_2 . Note that we divided the values of $ASR_{\text{electrode}}$ in Figure R2 by two since they are the symmetric cells. Especially, frequency analysis in Figure R2(c-d) shows the same trend with that of the single cell in Figure 2(g). The Ni-Rh cell exhibits slightly lower resistances than the REF cell at high frequencies under all fuel conditions due to the enhanced electrochemical activity for charge transfer at the triple-phase boundary (TPB) and the overall hydrogen oxidation reactions (HOR) at the fuel electrode. Under the CH_4 fuel condition, the medium frequency resistances significantly increase by a similar magnitude in both REF and Ni-Rh cells due

to the slow gas–solid interaction caused by the reduced partial pressure of H₂ and the sluggish CH₄ adsorption. However, although the low-frequency resistances for the REF cell significantly increase by sluggish gas reforming under CH₄ operation, those for the Ni-Rh cell almost remain unchanged. Therefore, it confirms that Ni-Rh slightly improves the electrochemical reaction for HOR under the H₂ environment and improves the gas reforming and gas-solid interaction under the CH₄/H₂O environment. Higher CH₄ conversion of the Ni-Rh fuel electrode at OCV condition measured by GC (Supplementary Fig. 9) further supports that Ni-Rh fuel electrode improves the sluggish gas–solid interaction and gas reforming.

We added the Figure R2 and related discussion in SI at Supplementary Fig. 8.

Figure R2. Nyquist plots of symmetric cell of REF and Ni-Rh fuel electrode under (a) H₂ and (b) CH₄/H₂O (S/C=1) environments at 500 °C. Area-specific polarization resistances according to different frequency ranges, high (> 10³ Hz), medium (10–10³ Hz), and low (< 10 Hz) frequencies under different fuel conditions (c) H₂, and (d) CH₄(S/C=1)).

Reviewer #3 (Remarks to the Author): In terms of the power density and durability, the results demonstrated in this work are good. However, there are a lot of experiments should be performed to clarify some results and strengthen the conclusions made in this work. The following concerns should be addressed before making the decision.

1. It is no clear if the Ni-Rh are bimetallic alloy. The TEM images are not clear and insufficient. Additional experiments should be performed to further confirm it.

We re-observed our samples under TEM and EDS and obtained clearer images, as shown in Figure 1 (below images). It shows the infiltrated Rh particles before the reduction process, which are identified solely from Ni. However, after the reduction process under H₂ atmosphere (Figure 1(c) and (e)), diffused Ni inside the BZCYYb lattice is subsequently exsolved and self-assembled with the infiltrated Rh forming the bimetallic alloy. It shows the socketed structure around the interface between exsolved particle and backbone structure, BZCYYb, which is the strong evidence of exsolution. Inside the particle, Rh is evenly dispersed (Figure R3 and Supplementary Fig. 3 in SI) which confirms the Ni-Rh bimetallic alloy formation. In addition, change in lattice constant in HR-TEM from $\sim 2.2 \text{ \AA}$ for Rh (111) to $\sim 2.15 \text{ \AA}$ for Ni-Rh (111) further verifies their bimetallic alloy formation.

[Before]

Fig. 1. Self-assembly process of the Ni-Rh bimetallic catalyst. (a) Schematics of the fuel electrode formation process for the REF cell (exsolved Ni particles) and Ni-Rh cell (self-assembly between infiltrated Rh and exsolved Ni particles). High magnification SEM images of the fuel electrode morphology for the (b) REF cell with exsolved Ni and (c) Ni-Rh cell with a self-assembled Ni-Rh bimetallic catalyst. Structure and chemical composition of the catalyst on the fuel electrode according to TEM and EDS mapping for the (d) REF and (e) Ni-Rh cells.

[After]

Fig. 1. Self-assembly process of the Ni-Rh bimetallic catalyst. (a) Schematics of the self-assembly process between infiltrated Rh and exsolved Ni particles. High magnification SEM images of the fuel electrode morphology of Ni-Rh cell (b) before reduction and (c) after reduction, respectively. Structure and chemical composition of the catalyst on the fuel electrode by TEM and EDS mapping with lattice spacing images for the Ni-Rh cell (d) before reduction and (e) after reduction, respectively.

Figure R3. Overlay image of EDS mapping between Ni and Rh confirming even distribution of Rh inside the Ni-Rh bimetallic alloy.

2. No TEM images of the anode prior to reduction were provided. It is unclear if the Rh is exsolved from the lattice.

We revised the Figure 1 as follow (Figure R4) to improve the reader's understanding of our fabrication process. As shown in Figure R4(a), Rh is firstly decorated onto the surface of the Ni-diffused BZCYYb backbone, and subsequently forms the Ni-Rh bimetallic alloy with exsolved Ni coming from the lattice of BZCYYb. Moreover, Figure R4(d) clearly shows the infiltrated-Rh particles before reduction solely identified from the lattice. In addition, the lattice constant of $\sim 2.2 \text{ \AA}$ for Rh (111) in HR-TEM further confirms its metallic phase.

Fig. R4. Self-assembly process of the Ni-Rh bimetallic catalyst. (a) Schematics of the self-assembly process between infiltrated Rh and exsolved Ni particles. High magnification SEM images of the fuel electrode morphology of Ni-Rh cell (b) before reduction and (c) after reduction, respectively. Structure and chemical composition of the catalyst on the fuel electrode by TEM and EDS mapping with lattice spacing images for the Ni-Rh cell (d) before reduction and (e) after reduction, respectively.

3. Figure 2a-2b, why the Ni-Rh PCFC performance under hydrogen was also significantly improved? Typically, the anode does not greatly affect the PCFC performance. Although figure 2g provides the ASRp, it is suggested to perform additional experiments and in-depth analysis to prove the performance improvement is due to the anode.

First of all, since other components such as electrolyte and cathode are all identical between REF and Ni-Rh cell except the Rh infiltration at the fuel electrode, we believe that the significant change in low frequency resistance in Figure 2(g) is largely attributed to the fuel electrode performance of Ni and Ni-Rh fuel electrode.

As the reviewer commented, the portion of anode in entire ASRp is typically low, especially under H₂. However, a number of previous studies show the enhancement of H₂ electrochemical reaction by adding the noble catalysts such as Pt, Ru, Pd, and Rh, which exhibit high electrochemical activities for hydrogen oxidation reaction (HOR) accompanied by H₂ adsorption and dissociation, simultaneously. (Tran et al. International Journal of Energy Research 45.4 (2021): 5325-5336., Kundu et al. Journal of Materials Chemistry A 6.46 (2018): 23531-23541., Durst et al. Journal of The Electrochemical Society 162.1 (2014): F190.) Thus, it can be inferred that the higher electrochemical performance of the Ni-Rh cell even under the H₂ environment is attributed to the improved H₂ dissociation and adsorption demonstrated by reduction of high and medium frequency resistances, respectively, as shown in Figure 2(g) under H₂.

Moreover, for in-depth analysis as the reviewer suggested, we conducted additional symmetric cell analysis of the REF and Ni-Rh fuel electrode under H₂ and CH₄/H₂O environments to further verify the effect of Ni-Rh on the decrease in ASRp, as shown in

Figure R5. Under the H₂ environment, the Ni-Rh cell shows the slight decrease in ASR_p (0.108 ohm·cm⁻² at 500 °C) compared to that of REF cell (0.140 ohm·cm⁻² at 500 °C). Under the CH₄/H₂O environment, ASR_p of the Ni-Rh cell shows a ~2-fold smaller value (0.249 ohm·cm⁻² at 500 °C) than that of the REF cell (0.565 ohm·cm⁻² at 500 °C), which is more significant difference compared to that under H₂. Note that we divided the values of ASR_p in Figure R5 by two since they are the symmetric cells. Especially, frequency analysis in Figure R5(c-d) shows the same trend with that of the single cell in Figure 2(g). The Ni-Rh cell exhibits slightly lower resistances than the REF cell at high frequencies under all fuel conditions due to the enhanced electrochemical activity for charge transfer at the triple-phase boundary (TPB) and the overall hydrogen oxidation reactions (HOR) at the fuel electrode. Under the CH₄ fuel condition, the medium frequency resistances significantly increase by a similar magnitude in both REF and Ni-Rh cells due to the slow gas–solid interaction caused by the reduced partial pressure of H₂ and the sluggish CH₄ adsorption. However, although the low-frequency resistances for the REF cell significantly increase by sluggish gas reforming under CH₄ operation, those for the Ni-Rh cell almost remain unchanged. Therefore, it confirms that Ni-Rh slightly improves the electrochemical reaction for HOR under the H₂ environment and improves the gas reforming and gas-solid interaction under the CH₄/H₂O environment. Higher CH₄ conversion of the Ni-Rh fuel electrode at OCV condition measured by GC (Supplementary Fig. 9) further supports that Ni-Rh fuel electrode improves the sluggish gas–solid interaction and gas reforming.

We added the Figure R5 and related discussion in SI at Supplementary Fig. 8.

Figure R5. Nyquist plots of symmetric cell of REF and Ni-Rh fuel electrode under (a) H₂ and (b) CH₄/H₂O (S/C=1) environments at 500 °C. Area-specific polarization resistances according to different frequency ranges, high (> 10³ Hz), medium (10–10³ Hz), and low (< 10 Hz) frequencies under different fuel conditions (c) H₂, and (d) CH₄(S/C=1).

Figure R6. Evaluation of CH₄ conversion at a fuel electrode at OCV condition in a single cell configuration.

4. Figure 2d, the IV curve under methane (s/c=2) is weird. It is suggested to retest it.

We re-performed the single cell test under methane operation (steam to carbon ratio of 2) and modified the figure 2 in the manuscript, as shown below.

[Before]

[After]

Fig. 2. Electrochemical performance evaluations of direct methane PCFCs. (a-d) Electrochemical performances of REF and Ni-Rh cells under different fuels (H₂, CH₄(S/C=2) and CH₄(S/C=1)) at 650 and 500 °C, where the fuel conditions are 97% H₂ with 3% H₂O for H₂ (100 sccm), 25% CH₄, 50% H₂O and 25% Ar for S/C=2 (32 sccm), and 25% CH₄, 25% H₂O and 50% Ar for S/C=1 (32 sccm), respectively. Air is fed into the cathode as an oxidant (100 sccm). Comparison of (e) the maximum power densities and (f) the area-specific polarization resistances with the previously reported PCFCs and SOFCs. (g) Area-specific polarization resistances according to different frequency ranges, high (> 10³ Hz), medium (10–10³ Hz), and low (< 10 Hz) frequencies, deconvoluted by DRT analysis under different fuel conditions (H₂, CH₄(S/C=2), and CH₄(S/C=1)).

5. The anode reforming activity is not evaluated in a packed bed reactor and compared with the reference anode. Otherwise, it is not clear to me if the performance improvement is due to the anode or something else. Therefore, anode reforming activity should be also evaluated in a packed bed reactor.

As the reviewer pointed out, an experiment with a powder-based packed bed reactor is a more general approach to compare the catalytic activity for CH₄ conversion. In our previous study (Hong et al., *Journal of Materials Chemistry A* 9.10 (2021): 6139-6151.), we have evaluated the steam reforming of methane supported by Ni/BZCYYb and Ni-Rh/BZCYYb in the powder-based packed bed reactor under various temperatures (Figure R7(a)), S/C ratios (Figure R7(b)), and space velocities (Figure R7(c)). As same as the results from the single cell configuration in this manuscript, the Ni-Rh bimetallic catalyst (Ni-Rh/BZCYYb) exhibits higher CH₄ conversion than the Ni monometallic catalyst (Ni/BZCYYb). In accordance with the packed bed reactor experiment, this study reveals that the Ni-Rh fuel electrode also exhibits the higher CH₄ conversion (Supplementary Fig. 9) compared to that of the REF fuel electrode even in a single cell configuration, leading to a higher electrochemical performance by sufficient H₂ supply. In addition, since other components such as electrolyte and cathode are all identical between REF and Ni-Rh cells, the significant change in thermo-catalytic performance for CH₄ conversion (Figure R8) and high electrochemical performance of PCFC (Figure 2) can be entirely attributed to the performance improvement of the Ni-Rh fuel electrode.

Figure R7. Evaluation of CH₄ conversion under packed bed reactor as a function of (a) temperature, (b) S/C ratio, and (c) space velocity. (Hong et al., Journal of Materials Chemistry A 9.10 (2021): 6139-6151.)

Figure R8. Evaluation of CH₄ conversion at a fuel electrode at OCV condition in a single cell configuration.

6. There is no evidence why the REF PCFC is unstable under methane. Is that due to coking or something else? More evidence should be provided.

As well as the carbon coking, there are other possible factors inducing the degradation such as segregation at the cathode, mechanical degradation of the electrolyte, chemical interdiffusion at the interface between electrode and electrolyte, Ni re-oxidation and etc. To clarify the contribution among them, we additionally evaluated the EIS analysis over long-term operation. As shown in Figure R9(a), ohmic and polarization resistances of the Ni-Rh cell remain almost unchanged, implying that there are no other significant factors of performance degradation in the Ni-Rh cell.

On the other hand, in Figure R9(b), the REF cell shows the significant increase in polarization resistance and slight increase in ohmic resistance. Except the Rh infiltration at the fuel electrode, since other components such as electrolyte and cathode are all identical between REF and Ni-Rh cells, we can conclude that severe degradation of the REF cell is mostly contributed to the degradation of the fuel electrode. The slight increase in ohmic resistance might be due to the loss of active sites in the fuel electrode. Moreover, as we compared in Figure 3(a), the CH₄ conversion of the REF cell decreases, showing a same trend with the decrease in electrochemical performance. On the other hand, the Ni-Rh cell maintains its CH₄ conversion as well as its electrochemical performance. It further confirms that the decrease in CH₄ conversion at the fuel electrode of the REF cell is a major factor of the decrease in the overall electrochemical performance.

In general, carbon-coking causes significant degradation of the catalyst, resulting in the deactivation of CH₄ conversion and losing their active sites. [Choi et al. *Nano Energy* 23 (2016): 161-171.] Thus, high tolerance of catalysts against carbon-coking is one of the

most critical issues for achieving the sustainability. [Yang, Lei, et al. "Enhanced sulfur and coking tolerance of a mixed ion conductor for SOFCs: BaZr_{0.1}Ce_{0.7}Y_{0.2-x}Yb_xO_{3-δ}." *Science* 326.5949 (2009): 126-129.] In accordance with the previous reports, we also found the clear evidence of carbon coking in the fuel electrode of the REF cell by EDS and Raman spectroscopy, as shown in Figure R10 (also in Figure 3(b) and Supplementary Fig. 14-15 in the SI). However, carbon coking did not appear in the Ni-Rh cell (The carbon cleaning mechanism is addressed in Figure 4-5 – we further conducted the In-situ DRIFTS to support the mechanism, which will be discussed in comment #7). Therefore, we can comprehensively conclude that degradation of electrochemical performance in the REF cell is predominantly attributed to the carbon coking in the fuel electrode. On the other hand, the Ni-Rh cell can maintain its stability with the high carbon coking tolerance.

We added the EIS analysis (Figure R9) over long-term operation to the Supplementary Fig. 13, and related discussion to lines 198-205 in the manuscript.

Figure R9. EIS analysis of (a) Ni-Rh cell and (b) REF cell over long-term performance evaluations operated with a fuel composition of 25% CH₄, 25% H₂O and 50% Ar at the fuel electrode with a total flow rate of 100 sccm and air at the cathode as an oxidant (100 sccm).

Figure R10. Carbon coking in a fuel electrode of REF cell verified by (a) EDS, and (b) Raman spectroscopy.

7. The proposed mechanism in Figure 4 looks pretty. However, there is a lack of evidence to support it. It is suggested to perform additional experiments or conduct computational modeling to support this mechanism.

As the reviewer pointed out, to support the evidence of carbon cleaning process, we further conducted the in-situ diffuse reflectance infrared Fourier transform spectroscopy (DRIFTS) experiments. For the carbon cleaning, reactions should follow the below pathways in Eq. (1) $C^* + O^* \rightarrow CO^*$, (2) $CH^* + O^* \rightarrow CHO^*$, and (3) $CH^* + OH^* \rightarrow CHOH^*$. Appearance of H₂O-related species (O* and OH*) and intermediate species (CO*, CHO* and CHOH*) can be evidence for verifying the carbon cleaning by in-situ DRIFTS measurements. [Socrates, George. Infrared and Raman characteristic group frequencies: tables and charts. John Wiley & Sons, 2004.] As shown in Figure 4, these surface species substantially appear in the Ni-Rh cell with peaks for CO*, CHO* and CHOH* in Supplementary Note 3, respectively. It implies the high H₂O dissociation and high carbon coking tolerance of the Ni-Rh bimetallic alloy. On the other hand, in the REF cell, they do not appear or shows substantially lower intensity than that of Ni-Rh, implying the low H₂O dissociation and low carbon coking tolerance.

Related to these findings from DRIFTS, we can support our results by using in-situ XPS. As we observed in DRIFTS, the Ni-Rh cell exhibits high H₂O-related species, which well matches with a large number of V_o and OH* that are the by-products of H₂O dissociation. Furthermore, we can conclude that C-Ni peaks in XPS, which represents the carbon coking in the Ni surface, disappeared, since Ni-Rh follows the reaction pathways in Eq. (1)-(3) as we verified by appearance of intermediate species (CO*, CHO* and CHOH*) in DRIFTS.

Based on these results, we added the DRIFTS results and related discussion as below to the Figure 4 and lines 227-241 in the manuscript and Supplementary note 3, respectively.

Fig. 4. Self-carbon cleaning mechanism on Ni-Rh bimetallic catalyst. (a) Schematic diagram of self-carbon cleaning on the Ni-Rh bimetallic catalyst. In-situ DRIFTS analysis at different wavenumber range of 1800–1200 cm^{-1} ((b) REF and (c) Ni-Rh cells) and 4000–3400 cm^{-1} ((d) REF and (e) Ni-Rh cells), respectively, during steam reforming of methane (3% CH_4 , 3% H_2O and 94% Ar for S/C=1, 20 sccm) in the temperature range of 100–500 $^{\circ}C$.

Since the CH₄ operation of S/C=1 at 500 °C is the thermodynamically favored regime for carbon-coking (Supplementary Fig. 8), high carbon-coking tolerance of the Ni-Rh cell implies the occurrence of self-carbon cleaning on the catalyst surface. The self-carbon cleaning process occurs through the following pathways (See Fig. 4(a)): 1) CO formation ($C^* + O^* \rightarrow CO^* + Ni^*$)⁴⁷, 2) CHO formation ($CH^* + O^* \rightarrow CHO^*$)⁴⁸, and 3) CHOH formation ($CH^* + OH^* \rightarrow CHOH^*$)⁴⁸ compared to the carbon-coking pathway, as shown in Supplementary Note 3. Therefore, we conducted the in-situ DRIFTS measurements to find the occurrence of the carbon cleaning process on the catalyst surface by the appearance of intermediate species of CO*, CHO*, and CHOH*⁴⁹⁻⁵¹, as shown in Fig. 4(b-c). In the Ni-Rh cell, representative peaks associated with CO* (1664 cm⁻¹), CHO* (1420–1370 cm⁻¹), and CHOH* (1440–1400 cm⁻¹) emerge as the temperature increases to 500 °C. On the other hand, in the REF cell, only CHO* species appears. In addition to observation of the formyl group, the carbon cleaning process is substantially accompanied by the evolution of hydroxyl species (3750–3550 cm⁻¹)^{52,53}, as shown in Fig. 4(d-e). The Ni-Rh cell exhibits a higher intensity of OH* than the REF cell. Moreover, the in-situ DRIFTS results show the increase in the formyl group as the temperature increases, accompanied by a simultaneous increase in hydroxyl species. Therefore, we can conclude that the Ni-Rh cell has the self-carbon cleaning process by generating more formyl group from the evolution of hydroxyl species, indicating that the Ni-Rh cell has higher carbon resistance than the REF cell. Detailed analysis of DRIFTS is explained in Supplementary Note 3 and Supplementary Fig. 17.

Supplementary Note 3.

Self-carbon cleaning mechanism by in-situ DRIFTS data measured on the fuel electrode surface

In-situ DRIFTS experiment was performed to define the species evolved on the fuel electrode surface of PCFCs during the steam reforming of methane (SRM) reaction, as shown in Supplementary Fig. 17. C=C stretching vibration (C-C bonding, 1515 cm^{-1}) and C-H deformation vibration (CH^* , 1340 cm^{-1}) related to carbon formation are common to REF and Ni-Rh, and their peak intensities in Ni-Rh are smaller than those in REF¹³. The results of carbon formation are consistent with the catalytic activity trends for long-term stability results. On the other hand, except for peaks related to carbon formation, other peaks such as formyl, methyl and hydroxyl are higher in Ni-Rh than REF. It can be seen that Ni-Rh is easier to form CO^* or CO_2^* than REF through the formation of C=O stretching mode (bridged-bonded CO_B^* , 1691 cm^{-1})¹⁴ and C-H deformation vibration (CHO^* , $1420\text{--}1370\text{ cm}^{-1}$)¹³. Ni-Rh is superior to REF for methyl (CH_X^*) dissociation through another C-H deformation vibration (CH_X^* , $1365\text{--}1295\text{ cm}^{-1}$)¹³ peak, which is equivalent to the catalytic activity trend related to methane reforming. Moreover, Ni-Rh has independent peaks such as CHO^* , CHOH^* , CO_T^* that do not present in REF. In Supplementary Fig. 17, the C=O stretching vibration (CHO^* , 1437 cm^{-1})¹⁴ peak is related to the C-H deformation vibration for CHOH^* ($1440\text{--}1400\text{ cm}^{-1}$)¹³, indicating that Ni-Rh forms a CHOH^* species by self-carbon cleaning unlike REF and leads to CHO^* . The C=O stretching mode (tilted CO_T^* , 1664 cm^{-1}) is formed on the surface containing Rh, which can form the other carbonyl species¹⁵. Furthermore, some peaks of Ni-Rh are greater than those of REF for CH_3 -metal groups due to CH_2 rocking vibration ($\text{CH}_3\text{-M}^*$, $900\text{--}700\text{ cm}^{-1}$), C-O stretching vibration (CO^* , $870\text{--}850\text{ cm}^{-1}$), and O-CO in-plane deformation vibration (COOH^* ,

675–590 cm^{-1})¹³. This indicates that Ni-Rh forms more COOH*, an intermediate species for CO or CO₂ production, than carbon formation from C* contained in CH₄. Therefore, the Ni-Rh bimetallic catalyst has higher carbon resistance than REF through the self-carbon cleaning process, as shown in Supplementary Fig.18.

Supplementary Figure 17. In-situ DRIFTS studies of the fuel electrode for self-carbon cleaning mechanism at (a) REF and (b) Ni-Rh, respectively, in the temperature range of 100–500 °C. The fuel conditions for the fuel electrode are 3% CH₄, 3% H₂O and 94% Ar for S/C=1 with the flow rate of 20 sccm.

Supplementary Figure 18. Expected reaction pathway of steam reforming of methane

for (a) REF and (b) Ni-Rh through in-situ DRIFTS analysis.

8. Additionally, there is lack of details provided in the paper/figures, making it hard to understand the PCFC testing conditions. For example, Figure 3, what's the current density?
There are a lot of similar issues.

We provided more details in the figure captions.

REVIEWER COMMENTS

Reviewer #1 (Remarks to the Author):

I think the authors have thoroughly improved the article based on my and the other reviewers' comments, and I think it is now acceptable for publication.

One of the other reviewers made the authors clarify that the cell is of merely 0.16 cm² area. That is very small, and such small active areas can sometimes lead to dominance of side-processes. I think it should be OK here as different catalysts are compared on the same area of electrodes, but if there is a further revision on other bases, I'd suggest the authors elaborate a bit more on the reasons and consequences of so small electrode areas.

Reviewer #2 (Remarks to the Author):

The revised manuscript addressed my concerns, therefore, I suggest accepting it for publication.

Reviewer #3 (Remarks to the Author):

With additional details provided to respond to the reviewers' comments, including the comments I have provided, there are a few critical issues, probably scientific errors, observed. These issues could greatly affect the innovation and impact of this work. These issues are listed as follows:

1. The active cathode area of the cell is only 0.16cm². The authors again did not provide the anode area. The anode could be big, which functions as the reforming catalyst, converting methane to a lot of H₂. The cathode is very small. So, the fuel cell performance is greatly overestimated. The results presented in this work is not solid. The authors should use a cathode active area of at least 0.5 cm² or bigger, depending on the anode area. Although previous publications report SOFCs or PCFCs with an active area of ~0.2 cm² or smaller, this does not indicate a small area is encouraged. We should avoid this.
2. Figure R4d, the authors claim the particles are Rh, which is wrong. The Rh cannot form metallic Rh after infiltration as their precursor is not metallic phase. It does not make sense that Rh and Ni can form alloy after reduction, which requires a good Ni diffusion into the Rh particles. There is no evidence to support this statement. Figure R4e cannot support this statement as the support also shows both Ni and Rh.
3. Figure r3 also has the same issue. The support also has both Ni and Rh. So, it is not right to claim the particles are alloyed particles.
4. To respond this question "Figure 2a-2b, why the Ni-Rh PCFC performance under hydrogen was also significantly improved? Typically, the anode does not greatly affect the PCFC performance. Although figure 2g provides the ASR_p, it is suggested to perform additional experiments and in-depth analysis to prove the performance improvement is due to the anode." The response cannot fully justify it. The authors should test the cell performance under H₂ and compare it with reference cells in a wide range of temperatures. By comparing the ASR_p as a function of temperature, additional information could be observed to support it.
5. The DRIFT results provided are not correct. If steam is fed to the DRIFT cell, there will be significant overlap with the key carbon-containing intermediates. Some intermediates, for

example, CHO cannot be detected. Additionally, some peaks in the current results are not clear, which cannot be correctly assigned.

REVIEWER COMMENTS

Reviewer #3 (Remarks to the Author):

With additional details provided to respond to the reviewers' comments, including the comments I have provided, there are a few critical issues, probably scientific errors, observed. These issues could greatly affect the innovation and impact of this work. These issues are listed as follows:

1. The active cathode area of the cell is only 0.16cm². The authors again did not provide the anode area. The anode could be big, which functions as the reforming catalyst, converting methane to a lot of H₂. The cathode is very small. So, the fuel cell performance is greatly overestimated. The results presented in this work is not solid. The authors should use a cathode active area of at least 0.5 cm² or bigger, depending on the anode area. Although previous publications report SOFCs or PCFCs with an active area of ~0.2 cm² or smaller, this does not indicate a small area is encouraged. We should avoid this.

We generally agree with the Reviewer's comments regarding the effective electrode size: the cell performance could be overestimated because lots of converted H₂ due to the large area for the reforming catalyst such as Ni in the anode could be supplied to the relatively small cathode. We used the cell in a size of 0.785 cm² (circle cell with a diameter of 1 cm) with a cathode size of 0.16 cm² (square with a length of 0.4 cm). The ratio of cathode area to anode area is ~20%. The cathode-to-anode-ratio of our cell is indeed relatively small compared to the commercial cells which generally have the cathode-to-anode-ratio of ~64% with the cell size of ~100 cm² and cathode area of ~64 cm². We added the information of the cell size in the manuscript (Line 273).

To avoid this issue, before confirming the operating condition, we firstly measured the methane conversion ratio in our cell at OCV and set the space velocity relatively high to show the lower methane conversion ratio in REF cell both in S/C=1 and 2 as shown in Figure R1 (Supplementary Fig. 9). Thus, REF cell is under lack of H₂ supply with $P(H_2)$ of ~40% at 600 °C and ~30% at 500 °C (Figure R2 and Supplementary Fig. 11). Through the results from Figure

R2, we additionally evaluated the electrochemical performances as a function of the $P(\text{H}_2)$ in Figure R3 (Supplementary Fig. 12). We can speculate that if H_2 supply in REF cell is already enough for the electrochemical reaction due to the relatively large anode reforming area (small cathode-to-anode-ratio), electrochemical performances would not be sensitive to the $P(\text{H}_2)$. However, Figure R3 and Supplementary Figure 6 show the linearly decreased maximum power density with the decrease in $P(\text{H}_2)$, confirming that the anode is under lack of fuel amount for the electrochemical reaction. Therefore, we believe that the overestimation of electrochemical performance in our cell due to the large portion of reforming catalyst in the anode is not significant to affect the overall performance. Furthermore, as the Reviewer #1 mentioned, since REF and Ni-Rh cell are compared fairly under the same condition, comparison in the lab-scale cell enables to focus on the material properties neglecting other factors such as gas diffusion, heat balance, and other operational parameters, which are hard to control in large-scale cell. In addition, many reports have used the small cathode-to-anode-ratio (0.15-0.4) in lab-scale cells as listed in Table R1 to just focus on the effect of structures and/or materials on the electrochemical performances without the above concerns.

In addition, as shown in Figure R4, we fabricated the large cell with a cathode size of $\sim 1.00 \text{ cm}^2$ and an anode size of $\sim 4.26 \text{ cm}^2$, and compared the electrochemical performances with the small cell under H_2 and CH_4 ($\text{S}/\text{C}=1$). Both REF and Ni-Rh cells demonstrate similar maximum power densities within $\sim 10\%$ difference compared to the smaller one as we used in this study (Table R2). Therefore, we believe that the cathode-or-anode-size effect would be very minor in our study.

Table R1. Cathode-to-anode-ratio in other reports.

No	Fuel Cell Type	Fuel	Cathode Area (cm ²)	Anode Area (cm ²)	Cathode-to-anode-ratio	Reference
This Work	PCFC	H ₂ , CH ₄	0.160	0.785	0.203	This Work
1	PCFC	H ₂ , CH ₄ , NH ₃	0.497	1.72	0.288	Duan et al., Nat. Energy, 2023
2	PCFC	H ₂ , CH ₄ , C ₂ H ₆	0.178	0.950	0.187	Ding et al., Nature, 604, 2022
3	PCFC	H ₂	0.2826	1.76	0.159	Park et al., Adv. Energy Mater., 13, 20232
4	PCFC	H ₂ , CH ₄	0.316	0.785	0.403	Liu et al., Nat. Energy, 3, 2018
5	PCFC	H ₂ , NH ₃	0.282	0.785	0.359	Liu et al., Energy Environ. Sci., 15, 2022
6	PCFC	H ₂ , NH ₃	0.431	1.327	0.324	Sullivan et al., Commun. Chem., 4, 2021
7	PCFC	H ₂ , Various Hydrocarbon fuels	0.5	1.76	0.282	Hayre et al., Nature, 557, 2018
8	PCFC	H ₂	0.2826	1.76	0.159	Park et al., Energy Environ. Sci., 2021

Figure R1. Catalytic activity of REF and Ni-Rh cells for methane steam reforming in the temperature range of 650-450 °C at OCV. CH₄ conversion and H₂ production rate of REF and Ni-Rh cells and their Arrhenius behaviors under (a-c) S/C=2 and (d-f) S/C=1. The fuel conditions for fuel electrode are 25% CH₄, 50% H₂O and 25% Ar for S/C=2, and 25% CH₄, 25% H₂O and 50% Ar for S/C=1, respectively, under same flow rate of 32 sccm.

Figure R2. Gas composition during cell operation under the CH₄-fueled operation (S/C=2, S/C=1) in the temperature range of 650-450 °C at OCV by gas chromatography. (a) raw data based on dry gas; and (b) calculated data including H₂O gas by C-H-O balance. The fuel conditions for fuel electrode are 25% CH₄, 50% H₂O and 25% Ar for S/C=2, and 25% CH₄, 25% H₂O and 50% Ar for S/C=1, respectively, under same flow rate of 32 sccm.

Figure R3. Comparison of maximum power densities for P_{H_2} and CH_4 fuel operation as a function of partial pressure of H_2 and temperatures (650–450 °C).

Figure R4. (a) Photos of the large cell, and (b) Electrochemical performance evaluations of large cell under H₂ and CH₄ (S/C=1).

Table R2. Performance comparison between small cell and large cell.

REF cell	MPD of small cell (In this manuscript)	MPD of large cell	MPD_large/MPD_small
H ₂ @ 650 °C	1.22	1.26	1.03
CH ₄ (S/C=1) @ 650 °C	0.560	0.530	0.946
H ₂ @ 500 °C	0.650	0.630	0.969
CH ₄ (S/C=1) @ 500 °C	0.160	0.151	0.943
Ni-Rh Cell	MPD of small cell (In this manuscript)	MPD of large cell	MPD_Small/MPD_Large
H ₂ @ 650 °C	1.47	1.46	0.993
CH ₄ (S/C=1) @ 650 °C	0.930	0.831	0.893
H ₂ @ 500 °C	0.690	0.707	1.02
CH ₄ (S/C=1) @ 500 °C	0.390	0.433	1.11

2. Figure R4d, the authors claim the particles are Rh, which is wrong. The Rh cannot form metallic Rh after infiltration as their precursor is not metallic phase. It does not make sense that Rh and Ni can form alloy after reduction, which requires a good Ni diffusion into the Rh particles. There is no evidence to support this statement. Figure R4e cannot support this statement as the support also shows both Ni and Rh.

Reflecting the Reviewer's comment, we re-checked the EDS mapping (Figure R5(a)) and further investigated the XRD patterns in Figure R5(b-d) to clearly verify the RhO formation before reduction and bimetallic alloy after reduction in Rh particles. In EDS mapping (Figure R4(a)), oxygen exists in Rh nanoparticles. In addition, as the reviewer pointed out, we found that Rh and RhO co-exist as partially oxidized Rh in the XRD patterns. We revised the manuscript as below and Supplementary Figure 3 adding the XRD patterns.

Moreover, as shown in Figure R5(c) and (d), Rh (111) and Rh (200) peak shifts toward the higher degree, confirming the change in the lattice parameters due to the alloy formation with Ni which has the smaller lattice constant. [Mozammel et al., Energy Fuels, 34, 2020] According to Zheng et al. Ni is a highly diffusible metal into the Rh. [Zheng et al., Acta Materialia 55, 2007] There has been a number of reports that have used the Ni-Rh bimetallic alloy under similar synthesis condition as shown in Table R3. Furthermore, the mechanism of bimetallic alloy formation accompanying the ex-solution (after reduction) has been well established in ex-solution society. [Kwon et al., J. Mater. Chem. A, 6, 2018] [Kwon et al., Nature Commun, 28, 2017] [Joo et al., Angew Chem, Int, Ed, 60, 2021] [Joo et al., Nature Commun, 11, 2019] [Joo et al., 6, Sci. Adv., 2020] [Lv et al., Nature Commun, 12, 2021] Therefore, we believe that Ni-Rh bimetallic formation after reduction is reasonable according to related previous reports and the experimental results that we verified in our samples using HR-TEM, EDS, and XRD.

- Line 99-102.

[Before]

In the Ni-Rh cell before reduction, Rh nanoparticles (4–8 nm) are decorated by infiltration on the BZCYYb surface. Rh nanoparticles exist solely as a metallic phase ($\sim 2.2 \text{ \AA}$ for Rh (111)²²) without mixing with Ni inside the BZCYYb lattice before reduction, as verified by HR-TEM image and EDS mapping in Fig. 1(d).

[After]

In the Ni-Rh cell before reduction, Rh nanoparticles (4–8 nm) are decorated by infiltration on the BZCYYb surface. XRD patterns and EDS mapping in Supplementary Figure 3 and Fig. 1(d) show that Rh nanoparticles exist solely as a partially oxidized metallic phase (RhO and Rh, $\sim 2.2 \text{ \AA}$ for Rh (111)²²) without mixing with Ni inside the BZCYYb lattice before reduction.

Figure R5. (a) EDS mapping of Ni-Rh cell before reduction, and (b-d) XRD patterns Ni-Rh cell before and after reduction.

Table R3. Conditions for Ni-Rh bimetallic alloy formation.

No.	Material	Synthesis Condition	Reference
This Work	Ni-Rh/BZCYYb	650 °C for 5h (After reduction)	This Work
1	Ni-Rh/CeO ₂ -Al ₂ O ₃	550 °C for 24h	Guo et al., ACS Catalysis, 1, 2011
2	Ni-Rh/CeO ₂ -Al ₂ O ₃	550 °C for 6h	Xie et al., Appl. Catal. A, General, 390, 2010
3	Ni-Rh/MgAlO	850 °C for 5h	Kim et al., Catal. Today, 136, 2008
4	Ni-Rh/La-Al ₂ O ₃	700 °C for 1h	Ferrandon et al., Appl. Catal. A, General, 379, 2010
5	Ni-Rh/CeO ₂ -Al ₂ O ₃	550 °C	Strohm et al., J. Catalysis, 238, 2006
6	Ni-Rh/Al ₂ O ₃	800 °C	Lakshapatri Catal. Sci. Technol., 3, 2013
7	Ni-Rh/CeO ₂ -ZrO ₂	750 °C for 4h	Horvath et al., Catal. Today, 169, 2011
8	Ni-Rh/BN	700 °C	Wu et al., Chem. Eng. J, 148, 2009
9	Ni-Rh/Al ₂ O ₃	700 °C	Wu et al., Chem. Eng. J, 148, 2009

[Modified Supplementary Figure 3 in the manuscript]

Supplementary Figure 3. (a) XRD patterns and EDS mapping of Ni-Rh cell (b) before and (c) after reduction.

XRD patterns and EDS mapping in Supplementary Figure 3 show that Rh nanoparticles exist solely as a partially oxidized metallic phase (RhO and Rh) without mixing with Ni inside the BZCYYb lattice before reduction. Moreover, as shown in Supplementary Fig 3(a), Rh (111) and Rh (200) peak shifts toward the higher degree after reduction, confirming the change in the lattice parameters due to the alloy formation with Ni which has the larger lattice constant.

3. Figure r3 also has the same issue. The support also has both Ni and Rh. So, it is not right to claim the particles are alloyed particles.

Same answer with Reviewer's comment #2

4. To respond this question "Figure 2a-2b, why the Ni-Rh PCFC performance under hydrogen was also significantly improved? Typically, the anode does not greatly affect the PCFC performance. Although figure 2g provides the ASRp, it is suggested to perform additional experiments and in-depth analysis to prove the performance improvement is due to the anode." The response cannot fully justify it. The authors should test the cell performance under H₂ and compare it with reference cells in a wide range of temperatures. By comparing the ASRp as a function of temperature, additional information could be observed to support it.

Reflecting the Reviewer's comment, we further evaluated the ASRp in a symmetric cell configuration under H₂ fuel as a function of temperature in a wide range (650-500 °C) in Figure R6. Ni-Rh cell shows the lower ASRp and lower E_a (0.91 eV for Ni and 0.86 eV for Ni-Rh), which confirms its high electrochemical performance even under H₂. It is due to the higher electrochemical activity of Ni-Rh for hydrogen oxidation reaction than that of Ni and uniform catalyst distribution of Ni-Rh bimetallic catalyst in a higher density, which can substantially enlarge the electrochemical reaction area.

We generally agree that the anode performance has not occupied a large portion of the overall fuel cell performance. However, Table R4 presents the recently published reports that highlight enhancement in overall electrochemical performance under H₂ fuel (maximum power density) by improvement in the anode performance. As shown in Figure R7, our improvement ratio is in a reasonable range similar to other reports. For the last decade, SOFC or PCFC society has primarily focused on improving the performance of cathode and electrolyte. However, the significant improvement in the ohmic resistance (electrolyte) and polarization resistance (cathode) concurrently has increased the proportion of the fuel electrode in the overall resistance. Therefore, we believe that the anode performance becomes more important nowadays, especially when utilizing non-hydrogen fuels such as methane and ammonia.

Figure R6. Nyquist plots of symmetric cell of (a) REF and (b) Ni-Rh fuel electrode under H₂. (c) Arrhenius plots of Area-specific polarization resistances.

Table R4. Performance enhancement under H₂ fuel by anode modification.

No.	Fuel Cell Type	Applied Catalyst	Temperature (°C)	MPD_Catalyst (mW/cm ²)	MPD_Bare (mW/cm ²)	Ratio (MPD_C/MPD_B)	REF
This Work	PCFC	Ni-Rh	650	1470	1220	1.20	This Work
		Ni-Rh	600	1130	960	1.17	
		Ni-Rh	550	880	790	1.11	
		Ni-Rh	500	690	650	1.05	
1	SOFC	Ru	800	500	400	1.25	Xiong et al., J. Mater. Sci. Tech., 125, 2022
2	SOFC	Ru	600	1500	1400	1.07	Thieu et al., Appl. Catal. B. Environ., 263, 2020
3	SOFC	Ru	500	120	75	1.60	Chen et al., Nat. Energy, 3, 2018
4	SOFC	Pd	700	300	180	1.67	McIntosh et al., Electrochem. Solid-state Lett., 2003
5	SOFC	Pt	700	610	310	1.97	Shin et al., Energy Environ. Sci., 15, 2022
6	PCFC	Pd	500	400	250	1.60	Park et al., J. Power Sources, 482, 2021
7	PCFC	Pd	600	1060	950	1.12	Jeong et al., Small, 2023
			550	830	710	1.17	
			500	550	470	1.17	
8	PCFC	Fe	700	1500	1450	1.03	Pan et al., Appl. Catal. B. Environ.,
			650	1150	1050	1.10	
			600	780	650	1.20	

							306, 2022
9	PCF C	Fe	700	2020	1860	1.09	Zhang et al., Energy Environ. Sci., 15, 2022

Figure R7. Performance enhancement under H₂ fuel by anode modification.

5. The DRIFT results provided are not correct. If steam is fed to the DRIFT cell, there will be significant overlap with the key carbon-containing intermediates. Some intermediates, for example, CHO cannot be detected. Additionally, some peaks in the current results are not clear, which cannot be correctly assigned.

As Reviewer referred, if a substantial amount of steam is fed to the DRIFTS cell, there might be significant overlap with the key carbon-containing intermediates, and some intermediates (e.g., CHO) might not be detected. However, in this study, the operating condition for steam that is fed into DRIFTS cell is only 3%. Thus, we could avoid this issue. In addition, there are previous reports that used DRIFTS under hydrocarbon reforming with steam condition to analyze the hydrocarbon steam reforming mechanism. As shown in Table R4, our operating conditions for steam partial pressure are suitable for investigating the mechanism for steam reforming of methane and are very similar to the range of other study conditions. In addition, as the Reviewer commented, some peaks in our results are not clearly identified. Therefore, we provided those peaks as a range, for example, CHO* (1420–1370 cm^{-1}), and CHOH* (1440–1400 cm^{-1}) and hydroxyl species (3750–3550 cm^{-1}) (Line 205-211). Nevertheless, the difference in peak evolutions of REF and Ni-Rh cell as a function of temperatures is clear, concluding the greater effect of Ni-Rh on the self-carbon cleaning.

Table R4. DRIFTS conditions under hydrocarbon reforming with steam condition to analyze the hydrocarbon steam reforming mechanism.

No.	Fuel	S/C ratio	P(H ₂ O) [%]	SCCM	REF
This work	Methane	1	3	20	This work
1	Methanol	33	90	138.75	Jacobs et al., Appl. Catal. A: Gen., 285.1-2, 2005
2	Ethanol	13	5.133268	20	Llorca et al., J. Catal., 227.2, 2004
3	Propane	3.333333	5	30	Kokka et al., Appl. Catal. B: Environ., 316, 2022
4	Ethanol	1.5	3	30	Sanchez et al., J. Phys. Chem. A., 114.11, 2010
5	Ethanol	4	8	10	Xu et al., ACS Catal., 3.5, 2013
6	Ethanol	3	-	20	Sharma et al., Int. J. Hydrog. Energy., 41.14, 2016
7	Ethanol	1.5	1.5	-	Kourtelesis et al., Appl. Catal. B: Environ., 284, 2021
8	Methanol	4	3.2	15	Frank et al., J. Catal., 246.1, 2007
9	Methane	2	3.1	65	Okada et al., Catal. Today, 307, 2018
10	Ethanol	1.5	-	30	Campos et al., Mol. Catal., 469, 2019

REVIEWERS' COMMENTS

Reviewer #3 (Remarks to the Author):

The authors have made great efforts to revising the manuscript. All my concerns have been addressed.